# FORGET-TO-FOCUS: CAN UNLEARNING IMPROVE DOMAIN SPECIALIZATION IN LLMS?

## ABSTRACT

Standard fine-tuning of Large Language Models for domain-specific tasks is often suboptimal due to interference from vast, pre-existing general knowledge from pretraining, leading to issues like negative knowledge transfer and the reinforcement of spurious correlations. We study whether removing parts of a pretrained model's pre-existing general knowledge before adaptation can make downstream learning easier. We propose and analyze *Forget-to-Focus*: a two-stage protocol that first performs targeted unlearning on a "forget" set (with an optional retain set for stability), then fine-tunes on a domain-specific dataset. Through rigorous experiments on different domains such as medical, mathematics, and coding benchmarks, we analyze whether this preparatory unlearning can lead to improved domain specialization. Our findings show that this protocol consistently outperforms standard fine-tuning e.g., it improves HumanEval pass@1 by 32.5% on Qwen3-0.6B and 11.95% on Qwen 72B model compared to standard fine-tuning. Beyond accuracy, we observe that F2F reshapes representational geometry as measured by centered kernel alignment, shifting models away from generalist initialization toward structures more conducive to in-domain specialization. Furthermore, unlearning prior fine-tuning helps improved calibration on medical QA tasks, reducing overconfidence and mitigating reliability issues that persist under standard fine-tuning. These findings establish unlearning not merely as a *privacy tool* but as a principled intervention for domain adaptation. By strategically suppressing irrelevant pretraining knowledge, *Forget-to-Focus* helps more stable optimization dynamics, better calibrated predictions, and consistently stronger downstream performance. The code is available at anonymous github : https://anonymous.4open.science/r/D-1545/README.md

## 1 INTRODUCTION

Fine-tuning large language models (LLMs) (Parthasarathy et al., 2024; Hu et al., 2022) on specialized target domains has shown impressive results, but it also comes with challenges of *negative transfer* (Zhang et al., 2022), where certain knowledge from vast, general-domain pre-training corpus actually hurts performance on the new domain specialized tasks. Pre-trained LLMs are exposed to vast general data, and when adapting them to a niche domain, they often carry over misleading correlations or behaviors that are *irrelevant* or even *conflicting* for the target domain. For example, a model fine-tuned for biomedical QA might still hold onto casual language patterns from web text that hinder learning precise medical terminology. Prior works (Sun & Dredze, 2025; Jiang et al., 2025) have shown that treating all pre-training knowledge as uniformly important prior is not optimal and some of that knowledge can degrade optimization and generalization on the target task or domain.

In other words, vanilla fine-tuning may struggle to "forget" irrelevant features, leading to slower convergence or suboptimal accuracy on in-domain data. This challenge motivates a shift in perspective. Thus, we ask a central research question: *Instead of passively hoping a model learns to ignore irrelevant knowledge, can we actively make it to forget this knowledge to enhance its capacity for new, specialized learning?* Notably Chen et al. (2023a) demonstrated that introducing an active forgetting mechanism during pre-training led to faster convergence and better low-resource adaptation to new languages.

This question leads us to the field of "Machine Unlearning"(Li et al., 2025), originally developed to address the "right to be forgotten" in response to data privacy regulations like GDPR Hoofnagle et al. (2019). "Machine unlearning" refers to algorithms that make a trained model intentionally 'forget' certain knowledge or data influences. Conventionally, unlearning has been studied for privacy (e.g., removing specific training examples from models upon request). In this work, we *repurpose* the concept of unlearning not for privacy, but to strategically remove or suppress irrelevant general knowledge that might hinder domain specialization.

However, leveraging unlearning for improved fine-tuning is not straightforward as (1) deciding what knowledge is harmful or useful is challenging, since the pretraining dataset is usually mixed with domain-irrelevant and domain relevant data, (2) unlearning aggressively could also erase general linguistic competence and useful information from the model, (3) optimization stability is uncertain in unlearning since it has potential to disrupt convergence and (4) it is unclear whether benefits extend across different domains and model scales (models with different architectures and sizes). These challenges motivate the need and an investigation of a protocol that carefully balances forgetting and retention to prepare models for effective specialization.

To address these challenges, we present *Forget-to-Focus (F2F)*, where we analyze if a preparatory unlearning phase can enhance the fine-tuning process. For this analysis, we employ a protocol where an unlearned model, created using a "forget set" of general data and a "retain set" for stability (depending upon the unlearning algorithm), is subsequently fine-tuned on a domain-specific dataset. We found that this preparatory unlearning consistently improves fine-tuning performance. Our experiments span multiple models with different architectures and sizes, and we investigate this phenomenon across the medical, mathematical, and coding domains and to deeply analyze why it occurs, we observe the change and shifts in model's internal representations.

Our contributions are as follows :

- We present the first comprehensive study of *machine unlearning* not as a privacy safeguard, but as a deliberate preparatory stage to enhance fine-tuning of large language models (LLMs) for domain specialization.

- We introduce *Forget-to-Focus* (F2F), a two-stage training procedure that strategically unlearns unnecessary general domain knowledge using a forget set (with an optional retain set), followed by domain-specific fine-tuning. This protocol consistently outperforms standard fine-tuning, DAPT, and parameter-efficient baselines across coding, mathematics, and medical domains.

- Through large-scale experiments on diverse models (from 0.6B to 72B parameters) with different architecture, we show that F2F helps in substantial pass@1 gains (e.g., $10.7\%$ performance increase on MBPP for Qwen-0.6B, and $9\%$ performance increase Qwen-72B compared to LoRA fine-tuning) while improving calibration on sensitive tasks such as medical QAs.

- Using centered kernel alignment (CKA), SVCCA, Fisher information, PCA-shift analyses, we observe that unlearning reshapes representational geometry, reallocates parameter sensitivity. These findings provide direct evidence that unlearning reduces negative transfer by suppressing interfering generalist features.

- Through extensive experiments, we show that both the size and quality of the forget set significantly impact fine-tuning performance, and that the relative weighting of the retain and forget sets further shapes performance across different models.

## 2  FORGET-TO-FOCUS

The current pattern of pre-training followed by fine-tuning leverages broad knowledge from large, general purpose datasets. However, this general knowledge is not always beneficial. When adapting a model to a specialized domain, a subset of the pre-trained knowledge can be irrelevant or even counterproductive, leading to a phenomenon known as negative transfer. We analyze that explicitly removing this irrelevant knowledge prior to fine-tuning allows the model to specialize more effectively. This leads to our central proposition.

Let $\theta_0$ denote pretrained parameters. We wish to specialize to domain $D$ with loss $L_D(\theta)$ while suppressing the rooted pretraining priors that induce negative transfer. The core intuition (Fig. 1) is that explicitly removing priors that can hurt the fine-tuning process, helps in a cleaner optimization landscape for specialization.

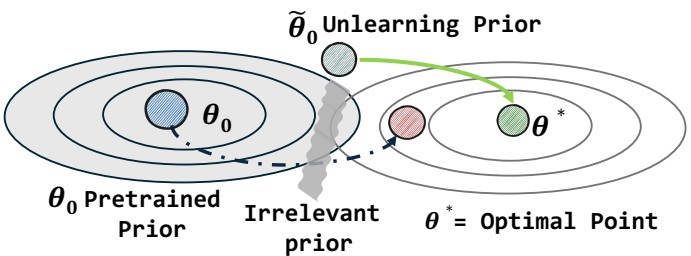

Figure 1: Schematic illustration of how pretraining priors create optimization barriers that slow convergence and induce suboptimal local minima when fine-tuning from $\theta_0$. Unlearning these priors yield a cleaner optimization path and lower final loss

Formally, let $\theta^\star = \arg\min_\theta L_D(\theta)$. The Forget-to-Focus (F2F) protocol constructs a new initialization $\tilde{\theta}_0$ such that

$$\|\tilde{\theta}_0 - \theta^\star\| < \|\theta_0 - \theta^\star\| \implies L_D(\text{FineTune}(\tilde{\theta}_0)) < L_D(\text{FineTune}(\theta_0)), \tag{1}$$

We assume access to (i) a *forget set* $F$ cause spurious general-domain behavior and (ii) a small *retain set* $R$ (often a subset of $D$) that preserves essential capabilities during unlearning. The objective we minimize to achieve Equation 1 is shown in the Equation 2 with gradient-accumulation averaging over $A$ micro-steps:

$$\tilde{\theta}_0 = \arg\min_\theta \frac{1}{A}\sum_{a=1}^{A}\left[-\lambda \underbrace{\ell_F^{(a)}(\theta)}_{\text{(GA/Forget term)}} + \sigma \underbrace{\ell_R^{(a)}(\theta)}_{\text{(GD/Retain term)}}\right], \tag{2}$$

where $\lambda, \sigma > 0$ weight the forget/retain terms. In practice we realize (2) via *gradient ascent* on $F$, forget set and *gradient descent* on $R$, retain set:

$$\theta \leftarrow \theta + \eta\,\lambda\,\nabla_\theta \ell_F(\theta) - \eta\,\sigma\,\nabla_\theta \ell_R(\theta), \tag{3}$$

where $\eta > 0$ is the step size. Thus features predictive on $F$ are de-emphasized or can be forgotten while $R$ stabilizes core competence. After $T_u$ steps we obtain $\tilde{\theta}_0$ and then fine-tune on $D$.

**Unlearn.** In this phase, we initialize $\theta \leftarrow \theta_0$. For $t=1{:}T_u$: sample minibatches $B_F \subset F$, $B_R \subset R$, compute $g_F = \nabla\ell(B_F; \theta)$, $g_R = \nabla\ell(B_R; \theta)$, and update via (3). Set $\tilde{\theta}_0 \leftarrow \theta$. **Retune.** And in retune phase, we initialize $\theta \leftarrow \tilde{\theta}_0$ and optimize $\min_\theta L_D(\theta)$ with standard fine-tuning.

While LLM training objective is non-convex, we use a convex linear surrogate to clarify the mechanism in a setting where the optimization is normal and interpretable. Consider, regularized linear models $f_\theta(x) = \theta^\top x$ with convex, $\beta$-smooth, $\mu$-strongly convex losses. Suppose the feature space decomposes as $\mathbb{R}^p = \mathcal{V} \oplus \mathcal{U}$, where $\mathcal{V}$ are *domain-relevant* directions and $\mathcal{U}$ are *irrelevant* (spurious) w.r.t. $D$. Assume $\theta^\star \in \mathcal{V}$ and the forget risk $L_F$ has curvature along $\mathcal{U}$ (Hessian $\succeq \mu_F I$ on $\mathcal{U}$).

**Proposition.** *(Contraction on irrelevant directions with bounded retain perturbation).* Considering the update in Equation 3 Assume: (i) the parameter space decomposes orthogonally as $\mathbb{R}^p = \mathcal{V} \oplus \mathcal{U}$ with projections $P_\mathcal{V}, P_\mathcal{U}$; (ii) $L_F$ is $\mu_F$ strongly convex on $\mathcal{U}$ (curvature lower bound $\mu_F > 0$ along $\mathcal{U}$); (iii) $L_F, L_R$ are $\beta$-smooth (gradient-Lipschitz with constant $\beta$); (iv) the retain gradient along $\mathcal{U}$ is uniformly bounded: $\|P_\mathcal{U}\nabla L_R(\theta)\| \leq G_R$ for all iterates. If $0 < \eta \leq 1/\beta$, then the $\mathcal{U}$ component contracts as

$$\|P_\mathcal{U}\theta^+\| \leq (1 - \eta\,\lambda\,\mu_F)\|P_\mathcal{U}\theta\| + \eta\,\sigma\,G_R.$$

Iterating for $T_u$ unlearn steps gives

$$\left\|P_\mathcal{U}\theta_{T_u}\right\| \leq (1 - \eta\,\lambda\,\mu_F)^{T_u}\left\|P_\mathcal{U}\theta_0\right\| + \frac{\sigma\,G_R}{\lambda\,\mu_F}.$$

Here, $\theta$ are model parameters; $\theta^+$ is the next iterate; $P_\mathcal{U}, P_\mathcal{V}$ are orthogonal projections onto the "irrelevant" subspace $\mathcal{U}$ and "relevant" subspace $\mathcal{V}$; $\mu_F$ is the strong convexity constant of $L_F$ along $\mathcal{U}$; $\beta$ is the smoothness constant; $G_R$ bounds the retain gradient on $\mathcal{U}$; $T_u$ is the number of unlearn steps.

**Corollary.** *(Retune convergence and downstream risk).* Let $\tilde{\theta}_0 := \theta_{T_u}$ be the post-unlearn iterate. Suppose the downstream objective $L_D$ is $\mu$ strongly convex and $\beta$-smooth with minimizer $\theta^\star \in \mathcal{V}$. Running gradient descent on $L_D$ with any step size $\alpha \in (0, 1/\beta]$ from $\tilde{\theta}_0$ satisfies:

$$T_{\mathrm{retune}}(\tilde{\theta}_0, \varepsilon) \;\leq\; \frac{\beta}{\mu} \, \log\Big(\frac{L_D(\tilde{\theta}_0) - L_D(\theta^\star)}{\varepsilon}\Big), \qquad L_D(\theta) - L_D(\theta^\star) \;\leq\; \frac{\beta}{2} \, \|\theta - \theta^\star\|^2.$$

Moreover, since $\theta^\star \in \mathcal{V}$,

$$\|\tilde{\theta}_0 - \theta^\star\| \;\leq\; \|P_\mathcal{V}\theta_0 - \theta^\star\| \;+\; (1 - \eta\,\lambda\,\mu_F)^{T_u} \|P_\mathcal{U}\theta_0\| \;+\; \frac{\sigma\,G_R}{\lambda\,\mu_F},$$

so increasing the forget to retain ratio $\lambda/\sigma$ tightens the starting distance for fine-tuning and hence improves both the iteration complexity and the final risk bound.

$\tilde{\theta}_0$ is the post unlearn initialization; $L_D$ is the downstream fine-tuning objective with smoothness $\beta$ and strong convexity $\mu$; $\theta^\star$ is its minimizer; $\varepsilon > 0$ is the target suboptimality; $T_{\mathrm{retune}}$ is the number of GD steps to reach $\varepsilon$.

# 3 EXPERIMENTAL SETUP

## 3.1 UNLEARNING ALGORITHMS

The F2F can be implemented using various machine unlearning algorithms. In practice, these methods realize the objective stated in Equation 2. In our experiments we explored the following unlearning methods :

(1) *GA+GD* : Using gradient ascent (GA) combined with gradient descent (GD) (Yao et al., 2024) on forget and the retain set *(GA+GD)* directly pushes the model parameters away from encoding the irrelevant data while simultaneously preserving the desired domain knowledge.

(2) *GA* $(\sigma = 0)$ : Using only gradient ascent on the forget set $(\sigma = 0)$. This is a more aggressive approach that focuses solely on forgetting, which can be effective if the retain set is not strictly necessary for maintaining core capabilities.

(3) *GA+KL* : Another approach is to use Kullback-Leibler divergence (KL) with GA to make sure the model does not diverge too much from the original parameters while preserving the desired domain-specific knowledge. In this case, the objective becomes $\tilde{\theta}_0 = \arg\min_\theta \; \frac{1}{A} \sum_{a=1}^A \Big[ - \lambda\, \ell_{\mathrm{F}}^{(a)}(\theta) \; + \; \sigma\,\mathrm{KL}(p_\theta \,\|\, p_{\theta_0}) \Big]$, where $p_{\theta_0}$ denotes the original model distribution.

(4) Negative Preference Optimization (NPO) (Zhang et al., 2024), samples from the forget set are treated as "unpreferred" or "rejected" responses. The model is then optimized to lower its likelihood of producing such outputs, effectively unlearning the associated knowledge while maintaining its general utility. The objective minimizes

$$\tilde{\theta} = \arg\min_\theta \; \mathbb{E}_{(x,y)\in\mathcal{F}} \left[ -\tfrac{2}{\beta} \log sigmoid\Big(-\beta \log \tfrac{\pi_\theta(y|x)}{\pi_{\mathrm{ref}}(y|x)}\Big) \right] \tag{4}$$

where $\pi_{\mathrm{ref}}$ is the reference model, and $\beta$ controls the sharpness of the penalty.

## 3.2 FINE-TUNING METHODS

To assess whether F2F consistently outperforms standard fine-tuning, we compare against following baselines:

(1) *SFT* (Supervised Fine-Tuning): Fine-tune all model parameters on task-labeled data with a cross-entropy objective, following the standard recipe (Devlin et al., 2019).

(2) *DAPT* (Domain-Adaptive Pretraining): Continue unsupervised pretraining on domain specific text prior to task-specific fine-tuning to better match target distribution (Gururangan et al., 2020).

(3) *LoRA*: Update only low-rank adapter matrices inserted into attention/FFN projections while freezing the original weights. This method reduces trainable parameters and memory both (Hu et al., 2022).

(4) *CurlLora* : We used (Fawi, 2024) which continually updates production LLMs with new data streams, minimizing model degradation and retraining costs.

## 3.3 MODELS AND DATASETS

We performed our experiments on models of different sizes, architecture, variants, and family : Qwen-2 72B-Instruct (Peng et al., 2023), LlaMA-2 13B (Touvron et al., 2023), LlaMA 3.1 8B-Instruct (Grattafiori et al., 2024), Gemma-2B-Instruct (Team et al., 2024) and Qwen-3-0.6B (Yang et al., 2025) to demonstrate the effectiveness of F2F to make the model adapt to certain domains. We conducted experiments across three domains: medicine, mathematics, and coding.

*Unlearning Step.*

For the unlearning step, we considered three types of forget sets from the Bookcorpus dataset (Kobayashi, 2018; Jagtap): (i) BC-select : a curated subset where we manually excluded texts overlapping with the target domain (e.g., biomedical for PubMedQA), focusing instead on general narrative and fiction content. This ensured that the forget set contained minimal domain-relevant knowledge, and (ii) BC-Mixed : a subset combining 800 random non-domain samples from BookCorpus with 200 domain-related samples (e.g., humaneval (Chen et al., 2021) for coding domain).(iii) BC-Cosine : a curated subset where we automatically extract samples which are not aligned with our target domain i.e., we encode each sample $x$ with a Transformer (Vera et al., 2025) $h_x = f_\theta(x)$, define the target-domain centroid $c_T = \frac{1}{|D_T|} \sum_{x' \in D_T} f_\theta(x')$, and rank samples by the cosine distance $d_{\cos}(x) = 1 - \frac{h_x^\top c_T}{\|h_x\| \|c_T\|}$, selecting samples with large $d_{\cos}(x)$. This setup interpolates between a clean forget set and one partially contaminated with target-domain knowledge. This helps us analyze of how domain overlap or forget set quality affects unlearning. The retain set is a small subset of the fine-tuning data, following prior work (Geng et al., 2025). Figure 2 demonstrates the clear boundary between the two domains in the BC-mixed dataset ensuring no domain leakage.

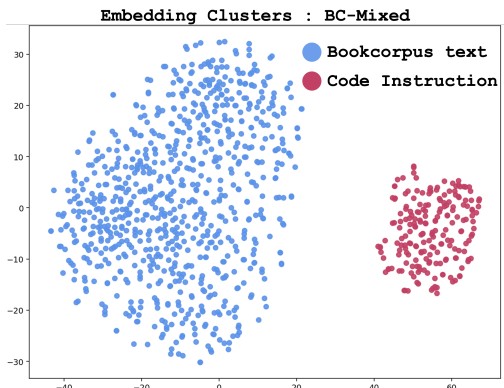

Figure 2: t-SNE of MiniLM (Wang et al., 2020) embeddings for the BC-Mixed forget set (800 BookCorpus non-domain + 200 domain-related code samples). The separation indicates distinct representational regions with limited overlap.

*Fine-tuning Step.* In the medical domain, we utilized PubMedQA (Jin et al., 2019), PubMed Guidelines (Chen et al., 2023b;c), and MedMCQA (Pal et al., 2022)'s training split as training datasets, and evaluated performance on the PubMedQA and MedMCQA test sets. For the coding domain, we trained on train set of OpenCoder (Huang et al., 2024) and evaluated on test set of HumanEval (Chen et al., 2021) and MBPP (Austin et al., 2021). For the mathematics domain, we used the NVIDIA's OpenMathInstruct-1 dataset (Toshniwal et al., 2024) for training, while evaluation was carried out on the Hendrycks MATH (Hendrycks et al., 2021) and GSM8K (Cobbe et al., 2021) benchmarks. For the evaluation, we used LM-EVALUATION-HARNESS repository by Eleuther (Gao et al., 2024b).

## 3.4 HYPERPARAMETER CONFIGURATION

*Unlearning Step.* During the unlearning step, we adopted a consistent set of hyperparameters across all models, unless otherwise specified. The base learning rate was fixed at $1 \times 10^{-5}$. We set the gradient ascent (GA) weight to 1.0 and the gradient descent (GD) weight to 0.5. The only exception was the LLaMA model, for which a higher learning rate of $3 \times 10^{-5}$ was found to be more effective in stabilizing convergence during unlearning. All models were trained with a batch size of *8* for Qwen 0.6B model and *2* for rest of the models. For the Qwen 72B model specifically, we employed QLoRA with rank 16 and a dropout 0.05, using bfloat16 precision.

*Fine-tuning Step.* For the fine-tuning stage, we set a uniform learning rate of $2 \times 10^{-5}$ across all models. Training epochs varied across models: the Qwen 0.6B model was finetuned for 8 epochs, while the remaining models were trained for a single epoch, due to their larger parameter sizes and to reduce the risk of overfitting on relatively small domain-specific datasets. All models were

optimized using the AdamW optimizer. The effective batch size was 128, obtained through gradient accumulation with an accumulation step size of 32. For Qwen-72B model, we adopt 4bit quantization and tuned it with only 50% of the original dataset.For models such as LLaMA-8B, LLaMA-13B, and Qwen-72B, we performed LoRA-based supervised fine-tuning (SFT) using FP16 precision. Trainings were performed on 80GB A100 GPUs.

## 4 EVALUATION

### 4.1 EFFECT OF F2F ON CODING PERFORMANCE

Table 1 presents pass@1 accuracies on MBPP and HumanEval for multiple model architectures and a comparative assessment of different fine-tuning strategies. For unlearning, we used 100 samples for Qwen-0.6B, and 1000 samples for the other models, with 1000 samples for the retain dataset. We can observe four principal insights :

(1) Across both Qwen, Gemma and LLaMA models, performing unlearning prior to fine-tuning yields consistently higher coding performance compared to fine-tuning alone. For Qwen 0.6B model, applying $Unl_{GA+GD}$ followed by fine-tuning improves performance on HumanEval from 19.50 to 42.07, demonstrating a considerable gain. Similarly, for LLaMA 8B-Instruct HumanEval performance increases by 22.7% after applying unlearning before fine-tuning compared with other fine-tuning methods, F2F enhances performance the most. These results confirm our central hypothesis that actively removing irrelevant pre-training knowledge can create additional capacity for specialized learning.

(2) Gradient ascent and descent combined ($Unl_{GA+GD}$) strategy consistently outperforms the GA-only variant. While GA-only unlearning sometimes leads to degradation or instability (e.g., LLaMA 8B HumanEval drops to 1.20 without subsequent fine-tuning), the GA+GD variant produces more reliable gains. This suggests that balancing removal (GA) with stability-preserving retention (GD) is crucial to prevent catastrophic forgetting of useful priors. However, the GA variant demonstrates that unlearning can be more effective than conventional fine-tuning approaches. For example, on the Qwen-0.6B model, GA achieves a pass@1 of 40.02 on HumanEval, surpassing LoRA (37.50) and SFT (31.71).

(3) The effect of unlearning varies across architectures. For models like Gemma 2B, unlearning had affected the performance (e.g., 0.00 performance after $Unl_{GA+GD}$). This indicates that aggressive unlearning may overwhelm models with limited capacity and limited pretraining domain specific knowledge. In contrast, after tuning, it improves the pass@1, and even performs better than usual fine-tuning where the model performance degrades after tuning.

4) Across fine-tuning methods, the performance challenge remains evident. In the case of the Gemma-2B-Instruct model, LoRA fine-tuning reduces HumanEval accuracy by 11.3%. However, following unlearning, performance improves substantially, rising by 29.4%.

((5) For Gemma-2B-Instruct, we observed that the strongest configuration is F2F+SFT, which slightly improves over the base model on MBPP and substantially improves HumanEval (Table 1); in contrast, the rows with large drops (e.g., $Unl_{GA+GD}$ without SFT) correspond to intermediate unlearning checkpoints rather than the final tuned models.

These observations highlight that preparatory unlearning causes more effective fine-tuning which strategically suppresses irrelevant pre-training knowledge causing the model align better with domain-specific objectives, thereby mitigating negative transfer and unlocking performance gains. Retention of broad skills beyond target domains are provided in Appendix A.

### 4.2 F2F w/ FINE-TUNING VARIANTS

To study the interaction between fine-tuning and unlearning, we tuned the models on a medical dataset and evaluated them under identical conditions. Table 2 highlights that across both model families, full SFT consistently delivers the strongest improvements, indicating that direct parameter updates provide the most effective alignment with domain-specific data. For Qwen 0.6B, SFT yields the largest gains, while LoRA and CurlLoRA provide modest but stable improvements, suggesting that lightweight adapters capture useful task knowledge but lack the depth of full tuning. DAPT

Table 1: MBPP and HumanEval pass@1 across different models (Qwen-2 72B-Instruct (Peng et al., 2023), LlaMA-2 13B (Touvron et al., 2023), LlaMA 3.1 8B-Instruct (Grattafiori et al., 2024), Gemma-2B-Instruct (Team et al., 2024) and Qwen-3-0.6B (Yang et al., 2025)) and tuning methods (higher is better). Best ; Second best

|  | | Qwen 0.6B | | Gemma 2B | | LlaMA 8B-Instruct | | LlaMA 13B | | Qwen 72B | |
|---|---|---|---|---|---|---|---|---|---|---|---|
| | Coding | MBPP | HumanEval | MBPP | HumanEval | MBPP | HumanEval | MBPP | HumanEval | MBPP | HumanEval |
| | (1) Base Model | 22.60 | 19.50 | **19.80** | 16.46 | 49.00 | 33.54 | 27.22 | 0.60 | 67.21 | 70.12 |
| | (1)+ $SFT$ | 28.80 | 31.71 | 12.80 | 16.20 | 56.60 | 56.71 | 37.01 | 40.21 | 69.50 | 71.12 |
| | (1) + $DAPT$ | 29.30 | 39.80 | 19.00 | 17.05 | 53.55 | 56.20 | 39.50 | 42.70 | **71.90** | 72.50 |
| | (1) + $LORA$ | 28.55 | 37.50 | 16.23 | 14.60 | 51.08 | 45.31 | 36.55 | 20.15 | 66.50 | 70.30 |
| | (1) + $CurlLora$ | 31.00 | 40.91 | 13.22 | 18.51 | 57.40 | 52.93 | 40.50 | 42.00 | 69.00 | 68.20 |
| F2F | (2) $Unl_{GA+GD}$ | 30.00 | 21.34 | 7.80 | 0.00 | 43.60 | 54.88 | 27.22 | 0.60 | 67.21 | 71.30 |
| | (2)+ $SFT$ | 31.60 | 42.07 | 20.05 | 21.30 | 60.10 | 60.37 | 50.31 | 46.15 | 72.50 | 78.50 |
| | (3) $Unl_{GA}$ | 24.00 | 20.73 | 0.80 | 0.00 | 22.60 | 1.20 | 0.00 | 25.50 | 60.05 | 65.02 |
| | (3)+ $SFT$ | 31.60 | 40.02 | 19.40 | 18.02 | 58.66 | 57.70 | 45.01 | 44.70 | 70.45 | 76.00 |

sits between the two, showing that continued pretraining transfers domain knowledge effectively but still underperforms SFT. For LLaMA 8B-Instruct, the pattern shifts: combining SFT with LoRA achieves the best balance of adaptation and efficiency, while LoRA and CurlLoRA trail behind, highlighting diminishing returns when adapters are applied in isolation. DAPT with LoRA provides gains but remains weaker than full SFT-based approaches, suggesting that structured fine-tuning remains essential for larger models.

Table 2: Evaluation results on PubMedQA and MedMCQA for Qwen 3 0.6B and LLaMA 3.1 8B-Instruct under different adaptation methods. ↑ Performance improvement over base model.

|  | Qwen 0.6B | | |  | LlaMA 8B-Instruct | |
|---|---|---|---|---|---|---|
| | PubMedQA | MedMCQA | | | PubMedQA | MedMCQA |
| SFT | **69.60**↑11.8 | 45.31↑13.06 | SFT | | **89.90**↑14.70 | **70.25**↑10.82 |
| LoRA | 64.35↑6.55 | 44.90↑12.65 | LoRA | | 85.00↑9.80 | 65.10↑5.67 |
| CurlLoRA | 65.00↑7.22 | 45.00↑12.75 | CurlLoRA | | 84.20↑9.00 | 63.40↑3.97 |
| DAPT | 68.00↑10.20 | 45.90↑13.45 | DAPT | | 88.65↑13.45 | 65.00↑5.57 |

## 4.3 F2F w/ Unlearning Variants

Figure 3 illustrates a comparative analysis in the medical domain (PubMedQA and MedMCQA) for two models of differing architectures and scales: Qwen-0.6B and LLaMA-8B-Instruct. The results show that combined gradient ascent and descent (GA+GD) unlearning yields the most substantial performance gains after fine-tuning, outperforming both unlearning-only and alternative unlearning approaches. Across PubMedQA and MedMCQA, unlearning reliably enhances the effectiveness of subsequent tuning. Notably, for smaller models such as Qwen-0.6B, tuning after $\sigma$=0 (only GA) unlearning tends to underperform, underscoring the importance of stability-preserving retention. In contrast, for larger models like LLaMA-8B, GA-only unlearning achieves performance comparable to, and in some cases exceeding, other unlearning variants due to the less dependency on stability-preserving corrections like GD.

## 4.4 Effect of Forget Set Quality

Table 3 compares performance when different forget sets (BC-Select vs. BC-Mixed vs. BC-Cosine) are applied across coding, medical, and mathematical domains. An important factor of F2F lies in the composition of the forget set. Across Qwen, Gemma, and LLaMA models, unlearning with a BC-Select forget set consistently produces greater downstream improvements following fine-tuning compared to using BC-Mixed. For instance, in the case of Qwen 0.6B, applying $Unl_{GA+GD}$ followed by tuning on BC-Select increases MBPP accuracy to 31.60, in contrast to 29.90 with BC-Mixed. This indicates that BC-Select, being more curated and less noisy, enables more precise removal of irrelevant pre-training features. Moreover, as it is not intermixed with domain-specific data points, it avoids erasing domain-relevant knowledge. In the case of BC-Cosine, where we selected forget set based on low cosine similarity demosntrates and proves to perform better than the baseline and other SOTA tuning methods. For LlaMA 8B-Instruct, the performance is very similar

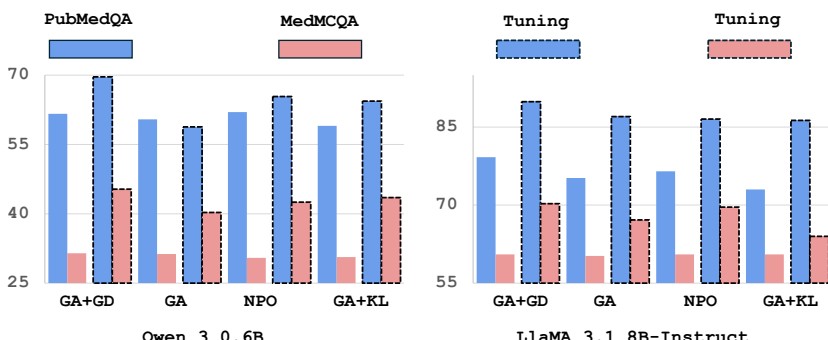

Figure 3: Comparative performance of different unlearning methods across different models with different architectures and sizes.

to the BC-Select which shows that cosine similarity can be used to select forget set if the domain is distinct.

Across all model sizes and domains, the results clearly demonstrate that the F2F protocol consistently outperforms standard fine-tuning. Models equipped with F2F show increasing gains in pass@1 accuracy. For instance, Qwen3-0.6B improves from 21.34 to 42.07 on HumanEval after applying unlearning and tuning, while LlaMA3.1-8B reaches 60.37 when compared to a baseline of 33.54.These improvements indicate that suppressing irrelevant pretraining knowledge helps models specialize in algorithmic reasoning.

These results highlight that the effectiveness of unlearning is highly dependent on the choice of forget set, the target domain, and capacity of the model. BC-Select forget sets appear more reliable for guiding domain adaptation, while BC-Mixed provides mixed benefits that depend on task alignment.

## 4.5 REPRESENTATION GEOMETRY ANALYSIS (CKA & SVCCA)

We analyze how unlearning Xu et al. (2025) and fine-tuning alter internal representations using *Centered Kernel Alignment* (CKA) Kornblith et al. (2019) and *Singular Vector Canonical Correlation Analysis* (SVCCA) Raghu et al. (2017).

**CKA .** Let $X \in \mathbb{R}^{N \times d_x}$ and $Y \in \mathbb{R}^{N \times d_y}$ be mean-pooled, sample-centered layer representations of the same inputs, i.e., $X_c = X - \bar{X}$ and $Y_c = Y - \bar{Y}$. The linear CKA is $\mathrm{CKA}(X,Y) = \frac{\|X_c^\top Y_c\|_F^2}{\|X_c^\top X_c\|_F \|Y_c^\top Y_c\|_F}$, which captures similarity of representational geometry and is invariant to orthogonal transforms and isotropic rescaling. Across the three domains, CKA reveals different representational drift patterns.Across all three domains, tuning overwrites most representations (low similarity), with F2F also highly divergent. This highlights that the extent of representational change depends on the domain, with F2F consistently pushing representations further from the unlearned model.

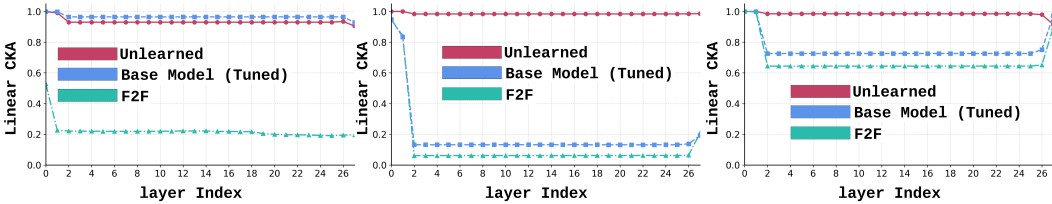

Figure 4: Representational drift measured by linear CKA. Across three domains, tuning substantially lowers similarity to the unlearned model; F2F exhibits the most pronounced departure. From left to right : Medical Domain, Mathematics Domain and Coding Domain.

Table 3: Effect of forget-set quality on F2F across domains. We compare curated *BC-Select* vs. mixed *BC-Mixed* vs. *BC-Cosine* forget sets on Qwen3-0.6B, LLaMA3.1-8B-Instruct, and LLaMA2-13B over coding (MBPP, HumanEval), medical (PubMedQA, MedMCQA), and math (Hendrycks-MATH, GSM8K) benchmarks (higher is better). **Best** ; Second best

| | FD | Method | Coding | | Medical | | Mathematics | |
|---|---|---|---|---|---|---|---|---|
| | | | MBPP | HumanEval | PubMedQA | MedMCQA | Hendrycks | GSM8K |
| Qwen3 0.6B | BC-Select | $(1)\ Unl_{GA+GD}$ | 30.00 | 21.34 | 61.60 | 31.44 | 39.07 | 0.26 |
| | | $(1) + Tuning$ | 31.60 | 42.07 | 69.60 | 45.31 | 54.11 | 15.30 |
| | | $(2)\ Unl_{GA}$ | 24.00 | 20.73 | 60.40 | 31.29 | 25.09 | 0.24 |
| | | $(2) + SFT$ | 31.60 | 40.02 | 58.80 | 40.26 | 51.20 | 14.00 |
| | BC-Mixed | $(1)\ Unl_{GA+GD}$ | 24.20 | 20.12 | 61.80 | 30.38 | 31.47 | 0.06 |
| | | $(1) + Tuning$ | 29.90 | 40.00 | 60.20 | 23.31 | 52.00 | 13.21 |
| | | $(2)\ Unl_{GA}$ | 23.80 | 20.12 | 60.20 | 31.89 | 25.00 | 0.00 |
| | | $(2) + Tuning$ | 28.00 | 33.10 | 61.20 | 35.43 | 50.00 | 13.51 |
| | BC-Cosine | $(1)\ Unl_{GA+GD}$ | 24.01 | 18.00 | 61.20 | 29.32 | 29.05 | 0.10 |
| | | $(1) + Tuning$ | 31.55 | 41.90 | 67.33 | 45.00 | 54.85 | 13.20 |
| | | $(2)\ Unl_{GA}$ | 22.22 | 18.23 | 58.45 | 29.00 | 25.00 | 0.00 |
| | | $(2) + Tuning$ | 29.91 | 40.50 | 57.65 | 38.75 | 48.90 | 13.07 |
| | - | $(3)\ Baseline$ | 22.60 | 19.50 | 57.80 | 32.25 | 41.09 | 0.02 |
| | | $(3) + Tuning$ | 28.80 | 31.71 | 62.60 | 42.12 | 50.06 | 12.30 |
| LLaMA3.1 8B | BC-Select | $(1)\ Unl_{GA+GD}$ | 43.60 | 54.88 | 79.21 | 60.50 | 18.00 | 61.70 |
| | | $(1) + Tuning$ | 60.10 | 60.37 | 89.90 | 70.25 | 29.50 | 70.51 |
| | | $(2)\ Unl_{GA}$ | 22.60 | 1.20 | 75.22 | 60.20 | 10.51 | 50.91 |
| | | $(2) + Tuning$ | 58.66 | 57.70 | 87.00 | 67.15 | 25.70 | 67.20 |
| | BC-Mixed | $(1)\ Unl_{GA+GD}$ | 40.50 | 52.52 | 79.50 | 59.30 | 17.00 | 51.00 |
| | | $(1) + Tuning$ | 56.20 | 55.76 | 87.61 | 70.10 | 28.81 | 65.20 |
| | | $(2)\ Unl_{GA}$ | 33.20 | 25.50 | 72.30 | 57.00 | 5.20 | 35.20 |
| | | $(2) + Tuning$ | 52.30 | 40.90 | 86.90 | 61.20 | 23.01 | 66.15 |
| | BC-Cosine | $(1)\ Unl_{GA+GD}$ | 42.55 | 53.76 | 79.00 | 58.22 | 17.91 | 61.00 |
| | | $(1) + Tuning$ | 59.55 | 59.86 | 85.31 | 71.02 | 28.33 | 68.57 |
| | | $(2)\ Unl_{GA}$ | 20.35 | 0.90 | 73.05 | 58.99 | 9.32 | 50.00 |
| | | $(2) + Tuning$ | 57.76 | 57.55 | 85.31 | 66.00 | 27.01 | 67.10 |
| | - | $(3)\ Baseline$ | 49.00 | 33.54 | 75.20 | 59.43 | 19.90 | 62.85 |
| | | $(3) + Tuning$ | 56.60 | 56.71 | 85.31 | 64.20 | 25.51 | 66.70 |
| LLaMA2 13B | BC-Select | $(1)\ Unl_{GA+GD}$ | 27.22 | 0.60 | 74.90 | 38.68 | 29.00 | 5.10 |
| | | $(1) + Tuning$ | 50.31 | 46.15 | 90.11 | 60.10 | 51.50 | 21.50 |
| | | $(2)\ Unl_{GA}$ | 0.00 | 25.50 | 70.00 | 36.51 | 24.35 | 2.00 |
| | | $(2) + Tuning$ | 45.01 | 44.70 | 89.33 | 57.93 | 50.90 | 20.00 |
| | BC-Mixed | $(1)\ Unl_{GA+GD}$ | 27.20 | 0.45 | 73.00 | 37.50 | 27.00 | 5.10 |
| | | $(1) + Tuning$ | 47.50 | 45.91 | 89.55 | 61.30 | 50.30 | 20.00 |
| | | $(2)\ Unl_{GA}$ | 0.00 | 10.00 | 65.99 | 29.55 | 23.55 | 1.05 |
| | | $(2) + Tuning$ | 39.55 | 40.01 | 87.00 | 50.60 | 47.60 | 20.00 |
| | BC-Cosine | $(1)\ Unl_{GA+GD}$ | 25.30 | 0.52 | 73.44 | 37.62 | 29.09 | 6.30 |
| | | $(1) + Tuning$ | 48.91 | 44.30 | 90.00 | 58.42 | 50.33 | 18.03 |
| | | $(2)\ Unl_{GA}$ | 0.00 | 23.33 | 69.55 | 36.22 | 24.00 | 0.92 |
| | | $(2) + Tuning$ | 44.33 | 42.05 | 89.10 | 60.99 | 49.62 | 19.52 |
| | - | $(3)\ Baseline$ | 27.22 | 0.60 | 75.20 | 38.68 | 27.5 | 2.00 |
| | | $(3) + Tuning$ | 37.01 | 40.21 | 86.30 | 54.00 | 37.09 | 16.30 |

**SVCCA .** SVCCA compresses each space to retain a fraction $\alpha = 0.99$ of variance via SVD, then computes the mean canonical correlation between the compressed features; if $X' \in \mathbb{R}^{N \times k_x}$, $Y' \in \mathbb{R}^{N \times k_y}$, and $\rho_1, \ldots, \rho_k$ are the canonical correlations with $k = \min(k_x, k_y)$, then $\text{SVCCA}(X, Y) = \frac{1}{k} \sum_{i=1}^{k} \rho_i$, emphasizing shared, high-variance factors which is complementary global geometry view of CKA. The SVCCA heatmaps indicate that tuning preserves partial alignment with the base model, while F2F introduces more substantial representational shifts. Base model vs. unlearned shows limited overlap beyond trivial self-similarity, whereas F2F vs. base model tuned reveals only localized correspondences. This highlights that tuning induces domain-dependent but structured drift, while F2F consistently drives stronger alterations in the representational subspace. *More analysis and ablations are given in the appendix section A.*

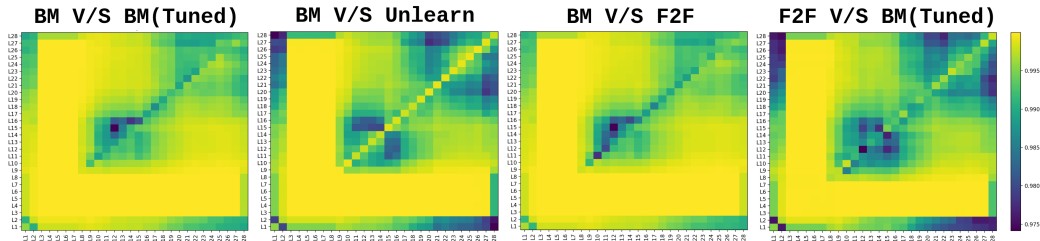

Figure 5: Heatmaps of representational similarity measured by SVCCA. We compare layer-wise representations across (i) Base model(BM) vs. fine-tuned base model, (ii) Base model (BM) vs. unlearned model, (iii) Base model (BM) vs. F2F, and (iv) Fine-tuned base model vs. F2F. SVCCA emphasizes alignment of shared high-variance factors.

## 5 CONCLUSION

We demonstrate that Forget-to-Focus (F2F) a simple two-stage pipeline : unlearns targeted general domain knowledge (forget) and then fine-tunes to adapt to a domain specific model (focus) consistently improves domain adaptation of LLMs across coding, math, and medical tasks, from 0.6B to 72B scales. F2F delivers higher accuracy than standard fine-tuning and parameter-efficient baselines, improves calibration on sensitive QA, and induces clear representational shifts (via CKA/SVCCA, Fisher, PCA) away from generalist features toward domain-useful structure. These gains arise from suppressing interfering priors from pretraining, causing stabler optimization and reduced spurious correlations. The method is modular, data-driven (via forget/retain sets), and compatible with common training stacks. Overall, F2F reframes unlearning as capacity reallocation for specialization, offering a practical path to more reliable in-domain LLMs.

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

# A APPENDIX FOR FORGET-TO-FOCUS

## A.1 MBPP AND HUMANEVAL PASS@1 ACROSS DIFFERENT VARIANTS OF QWEN 2.5 FAMILY.

As extension of Table 1, Table 4 presents pass@1 accuracies on MBPP and HumanEval for Qwen 2.5 model variants and a comparative assessment of different fine-tuning strategies. In particular, $Unl_{GA+GD}$ pretraining followed by SFT (2)+SFT results the strongest performance for every model size, achieving new best pass@1 scores of 53.80 & 47.30 (MBPP/HumanEval) for 1.5B, 75.90 & 53.20 for 7B, and 71.30 & 51.59 for 14B. Compared to the respective base models, this corresponds to absolute gains of +13.8, +10.1, and +9.1 points on MBPP and +11.9, +13.0, and +14.8 points on HumanEval for the 1.5B, 7B, and 14B models, respectively, which translate to roughly 30 - 40% relative improvements.

Overall, these results demonstrate that leveraging F2F is a robust and scalable strategy for enhancing code generation capabilities across model sizes of the same famnily.

Table 4: MBPP and HumanEval pass@1 across Qwen 2.5 model variants and tuning methods (higher is better). **Best** ; Second best

| Coding | Qwen 2.5 1.5B | | Qwen 2.5 7B | | Qwen 2.5 14B | |
|---|---|---|---|---|---|---|
| | MBPP | HumanEval | MBPP | HumanEval | MBPP | HumanEval |
| (1) Base Model | 40.00 | 35.37 | 65.85 | 40.20 | 62.20 | 36.80 |
| (1)+ SFT | 45.04 | 38.25 | 72.53 | 43.80 | 65.75 | 40.55 |
| (1)+ DAPT | 46.00 | 41.03 | 71.35 | 44.65 | 69.00 | 41.69 |
| (1)+ LORA | 44.76 | 39.01 | 70.33 | 44.83 | 67.35 | 49.97 |
| (1)+ CurlLora | 46.22 | 43.21 | 72.00 | 45.09 | 68.00 | 41.33 |
| (2) $Unl_{GA+GD}$ | 43.00 | 37.11 | 65.60 | 45.10 | 64.55 | 40.25 |
| (2)+ SFT | **53.80** | **47.30** | **75.90** | **53.20** | **71.30** | **51.59** |
| (3) $Unl_{GA}$ | 39.61 | 34.30 | 67.02 | 43.21 | 61.11 | 37.81 |
| (3)+ SFT | **52.40** | **45.80** | 72.50 | **48.70** | 68.45 | 48.10 |

## A.2 RETENTION OF BROAD SKILLS BEYOND TARGET DOMAINS

We evaluated broad-skill retention across ARC-E/C Clark et al. (2018), HellaSwag (Zellers et al., 2019), Winogrande (Sakaguchi et al., 2021), PIQA (Bisk et al., 2020), and BoolQ (Clark et al., 2019) for Qwen-0.6B and LLaMA-8B with different fine-tuning settings. In Table 5, we observe that simple supervised fine-tuning improves tasks like ARC and BoolQ, while often reducing performance on commonsense benchmarks such as HellaSwag, Winogrande, and PIQA, indicating a trade-off between specialization and everyday reasoning. Unlearning with gradient ascent plus gradient descent produces small but consistent gains across most tasks with minimal regressions, suggesting a stabilizing effect on broad skills. The full Forget-to-Focus pipeline that combines GA+GD unlearning followed by supervised fine-tuning strengthens knowledge retention further and largely preserves commonsense accuracy, yielding a near Pareto improvement relative to the base model in many cases. In contrast, gradient-ascent-only unlearning is volatile, with large swings across datasets; applying supervised fine-tuning afterward recovers much of the instability yet still leaves task-selective regressions, particularly on PIQA for the smaller and mid-size settings. Taken together, these trends support the claim that the proposed unlearn-then-retune recipe can retain broad capabilities while enabling targeted forgetting, provided the unlearning stage includes an explicit retain mechanism rather than relying on ascent alone.

To assess conversational and instruction-following robustness, we additionally evaluated models on Alpaca-Eval using the length-controlled win rate and the official lmsys-gpt4 annotator configuration. We find that F2F slightly improves win rate compared to the base, indicating that the proposed unlearning–retuning procedure reallocates capacity toward domain specialization without sacrificing instruction-following or conversational fluency. Together, these results reinforce that F2F enables targeted forgetting while preserving general and interactive capabilities.

Table 5: Broad-skill retention audit across general benchmarks (accuracy) and Length Controlled Win rate for Alpaca-Eval.

| Model | Method | ARC-E | ARC-C | HellaSwag | Winogrande | PIQA | BoolQ | Alpaca-Eval (LC Win Rate) |
|---|---|---|---|---|---|---|---|---|
| Qwen 0.6B | (1) Base Model | 68.00 | 32.50 | 45.00 | 59.00 | 67.50 | 67.50 | 28.58 |
| | (1) + $SFT$ | 68.50↑0.5 | 36.50↑4.0 | 44.50↓0.5 | 58.50↓0.5 | 65.00↓2.5 | 71.00↑3.5 | 27.65↓0.9 |
| | (2) $Unl_{GA+GD}$ | 67.50↓0.5 | 36.00↑3.5 | 45.50↑0.5 | 61.00↑2.0 | 68.00↑0.5 | 72.00↑4.5 | 28.38↓0.2 |
| | (2) + SFT | 68.50↑0.5 | 37.00↑4.5 | 45.00(0) | 57.50↓1.5 | 64.00↓3.5 | 73.50↑6.0 | 29.01↑0.4 |
| | (3) $Unl_{GA}$ | 67.50↓0.5 | 32.50(0) | 59.00↑14.0 | 44.50↓14.5 | 66.50↓1.0 | 75.00↑7.5 | 29.59↑1.1 |
| | (3) + SFT | 69.00↑1.0 | 35.00↑2.5 | 45.00(0) | 63.50↑4.5 | 58.50↓9.0 | 73.50↑6.0 | 27.78↓0.7 |
| LLaMA 8B | (1) Base Model | 82.50 | 52.50 | 55.00 | 80.00 | 79.50 | 87.50 | 30.22 |
| | (1) + $SFT$ | 83.00↑0.5 | 56.50↑4.0 | 51.50↓3.5 | 74.30↓5.7 | 76.00↓3.5 | 89.00↑1.5 | 30.32↑0.1 |
| | (2) $Unl_{GA+GD}$ | 82.00↓0.5 | 56.00↑3.5 | 55.50↑0.5 | 82.00↑2.0 | 80.50↑1.0 | 90.50↑3.0 | 30.22(0.0) |
| | (2) + SFT | 83.00↑0.5 | 59.00↑6.5 | 55.00(0) | 79.00↓1.0 | 78.50↓1.0 | 91.00↑3.5 | 30.55↑0.3 |
| | (3) $Unl_{GA}$ | 82.00↓0.5 | 52.50(0) | 69.50↑14.5 | 65.00↓15.0 | 78.50↓1.0 | 86.50↓1.0 | 30.12↓0.1 |
| | (3) + SFT | 84.00↑1.5 | 55.00↑2.5 | 68.00↑13.0 | 81.00↑1.0 | 77.00↓2.5 | 90.00↑2.5 | 30.42↑0.2 |

Table 6: Forgetting verification across models. $\Delta\text{NLL} = \text{NLL}_C - \text{NLL}_O$; higher means more forgetting.

| Model | Mean $\Delta$NLL | 95% CI | Median | Cohen's $d$ |
|---|---|---|---|---|
| **Base Model (Tuned)** | **0.347** | $[0.234, 0.472]$ | 0.297 | 1.72 |
| **Unlearn** | **0.681** | $[0.559, 0.831]$ | 0.563 | 2.90 |
| **F2F** | **0.678** | $[0.566, 0.819]$ | 0.609 | 3.07 |

### A.3 PROBING VERIFICATION / FORGOTTENNESS

Because the forget set $\mathcal{D}_f$ does not target a single domain, we verify forgetting via a probing methodology with sparse autoencoders (SAEs). We train sparse-coders for the base model $O$ and the comparison model $C$ using EleutherAI's SPARSIFY framework Gao et al. (2024a); EleutherAI (2024), on the final layer representation of Qwen3-0.6B and its two other variants : F2F, and unlearned only with BookCorpus. For each $x \in \mathcal{D}_f$, we compute the per-example difference $\Delta\text{NLL}(x) = \text{NLL}_C(x) - \text{NLL}_O(x)$, so larger values indicate that $C$ assigns lower likelihood than $O$ i.e., greater "forgottenness" of the targeted content. We summarize the distribution of $\Delta$NLL with its mean, a 95% percentile bootstrap CI over the mean ($B=2000$ resamples of examples), the median, and a standardized effect size (Cohen's $d$; one-sample against 0, using the sample standard deviation of $\Delta$NLL) which are reported in Table 6. Unlearn and F2F exhibit the largest forgetting on $\mathcal{D}_f$, with mean $\Delta$NLL of 0.681 $[0.559, 0.831]$ and 0.678 $[0.566, 0.819]$, respectively. Although their means are essentially tied, F2F shows a higher median (0.609 vs. 0.563) and a larger effect size (Cohen's $d = 3.07$ vs. 2.90), indicating a more uniformly strong shift across examples. In contrast, the Base Model (Tuned) also shows forgetting but at substantially smaller magnitude (mean 0.347, CI $[0.234, 0.472]$, median 0.297, $d = 1.72$). Overall, the table supports the conclusion that targeted procedures (Unlearn/F2F) induce considerably greater forgottenness than generic fine-tuning, with F2F displaying slightly stronger concentration of the effect.

### A.4 PCA-SHIFT

To better understand how different training interventions alter internal representations (Xu et al., 2025), we performed a layer-wise PCA shift analysis. For each layer $L$, we extracted the mean hidden representations of a shared set of prompts from both the reference model and its variants, fit a PCA on the reference representations, and projected all models into this space. The displacement along the first principal component was computed as $\Delta\text{PC1}(L) = \overline{\mathbf{h}}_{\text{model}}^{(L)} \cdot \mathbf{u}1 - \overline{\mathbf{h}}\text{ref}^{(L)} \cdot \mathbf{u}_1$, where $\mathbf{u}_1$ is the top PCA direction of the reference and $\overline{\mathbf{h}}^{(L)}$ is the mean hidden state. This analysis reveals that Base model-tuned models exhibit broad and uniform representational drift across many layers, indicating a global reshaping of internal geometry. In contrast, the Unlearned model remains closely aligned with the base model, with only minimal deviations concentrated in a few higher layers. Strikingly, F2F induces shifts of comparable magnitude to Baseline-Tuning but in a far more targeted manner, selectively altering specific layers while preserving much of the base geometry. This suggests that F2F achieves efficient and precise representational adaptation, retaining general capabilities while reallocating capacity only where necessary.

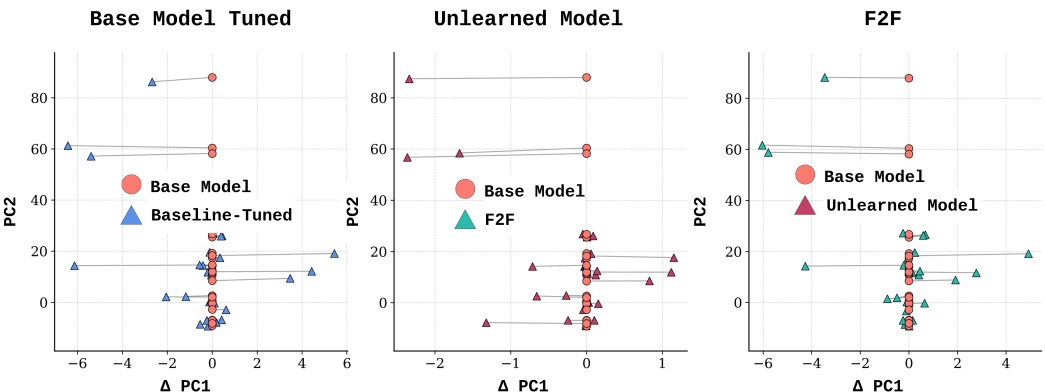

Figure 6: PCA shift analysis shows Baseline-Tuned models drift broadly, Unlearned models stay close to the base, and F2F induces targeted shifts that balance adaptation with stability.

## A.5 FISHER'S ANALYSIS

To understand how unlearning redistributes parameter sensitivity, we analyze the empirical Fisher information (Cha et al., 2024) of attention projections (Figure 7). For each block $l$ and role $r \in \{q, k, v\}$, we estimate the diagonal Fisher as $\widehat{F}_{l,r} = \frac{1}{B} \sum_{b=1}^{B} \left( \nabla_{W_r^{(l)}} \ell(x_b; \theta) \right)^2$, where $\ell(x; \theta)$ is the token-level NLL. Head-wise values are obtained by slicing the row dimension of $W_r^{(l)}$ into $H$ heads of size $d = \text{hidden\_size}/H$, averaging within each slice, and then across roles: $f_h^{(l)} = \frac{1}{3} \sum_{r \in \{q,k,v\}} \frac{1}{|S_h|} \sum_{(i,j) \in S_h} \widehat{F}_{l,r}[i,j]$ We report the median and interquartile range of $\{f_h^{(l)}\}_{h=1}^{H}$ within each block to capture depth-wise sensitivity. Figure 7 shows that standard fine-tuning sharply amplifies Fisher values in shallow layers, reflecting unstable reliance on low-level pretrained features. F2F with $\sigma = 0.5$ dampens shallow-layer sensitivity while retaining moderate activity across depth, striking a balance between stability and useful priors. In contrast, $\sigma = 0$ further suppresses Fisher values and yields smoother, more uniform profiles, favoring robustness and calibration but with reduced representational leverage. Overall, F2F systematically reshapes the sensitivity landscape to enable more stable and domain-aligned specialization.

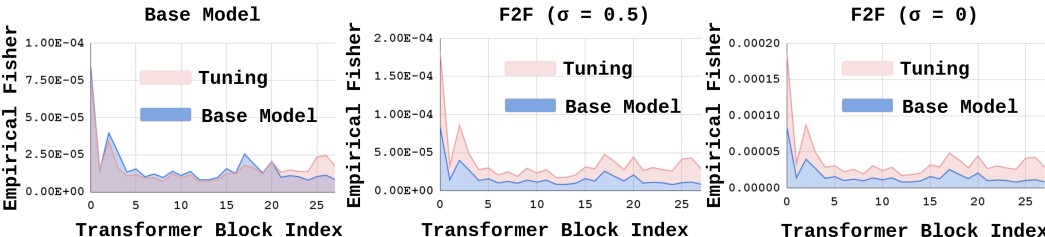

Figure 7: Head-wise redistribution of attention sensitivity after unlearning, measured by empirical Fisher information.

## A.6 CALIBRATION & RISK COVERAGE

To study how unlearning and fine-tuning affect calibration, we use *reliability diagrams* and *confidence histograms*. Given logits $z \in \mathbb{R}^C$ and labels $y$, we compute probabilities $p = \text{softmax}(z)$, confidence $c_i = \max_j p_{i,j}$, and correctness $\mathbb{1}[\arg\max_j p_{i,j} = y_i]$. Grouping confidences into $M$ bins $\{I_m\}$, average confidence $\hat{c}_m$ and accuracy $\hat{a}_m$ yield a reliability curve $(\hat{c}_m, \hat{a}_m)$ against the diagonal $y = x$, while Expected Calibration Error (ECE) measures deviation. Confidence histograms visualize the spread of predictions across $[0, 1]$. Calibration is critical in medical QA: overconfident errors are harmful, while underconfident correct answers reduce utility. From Figure 8, we observe that in F2F protocol, unlearning alone increases uncertainty, however, after fine-tuning the model

recovers a well calibrated profile whose reliability curve follows $y = x$ and whose confidences are broadly distributed. In contrast, baseline fine-tuning collapses confidences around 0.3-0.5, producing poorly calibrated outputs. Overall, *F2F achieves better calibration than fine-tuning alone*, improving both trustworthiness and domain adaptation.

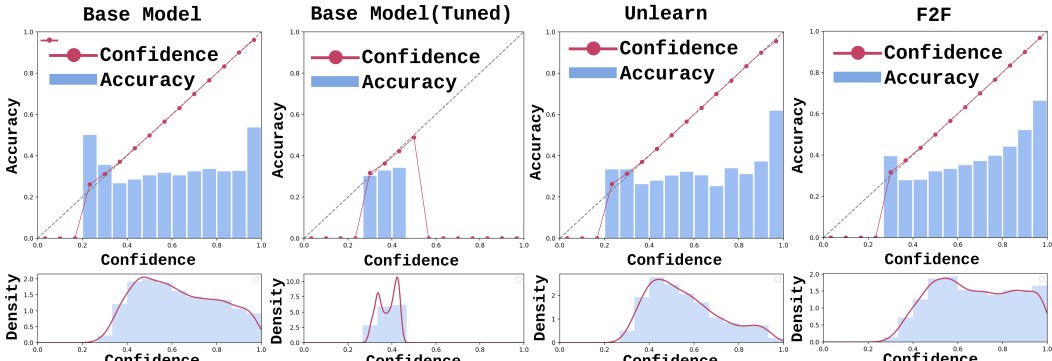

Figure 8: Reliability diagrams and confidence distributions on MedMCQA. F2F produces better-calibrated probabilities than fine-tuned base model.

Table 7: Calibration and likelihood metrics on MedMCQA ($\downarrow$ lower is better).

| Model Types | NLL$\downarrow$ | Brier$\downarrow$ | ECE$\downarrow$ |
|---|---|---|---|
| Base Model | 1.851 | 0.911 | 0.308 |
| Base Model (Tuned) | 1.762 | 0.825 | 0.277 |
| **F2F** | **1.392** | **0.751** | **0.050** |
| Unlearning | 1.659 | 0.867 | 0.256 |

### A.7 SPECTRAL SURROGATE ANALYSIS FOR LORA CAPACITY

We introduce a *spectral surrogate analysis* to estimate the intrinsic rank capacity required for LoRA without full fine-tuning. The method instruments LoRA-targeted linear layers (attention projections $q/k/v/o$ and MLP up/down/gate projections) to collect activations $A$ and output gradients $G$ on a small calibration set. From these, we construct the cross-covariance matrix $C = YX^\top/N$ with $X = A^\top$ and $Y = G^\top$. A randomized SVD yields singular values $\{s_i\}$, which define the cumulative explained variance curve $\text{EV}(r) = \frac{\sum_{i=1}^r s_i^2}{\sum_i s_i^2}$. We summarize model capacity using two aggregate measures: (i) an *energy-weighted average curve* across layers, reflecting overall compressibility, and (ii) a *layerwise minimum curve*, which highlights bottleneck layers that require higher ranks. By sweeping $r$, we obtain intrinsic rank estimates (e.g., the smallest $r$ such that $\text{EV}(r) \geq 0.9$), identify non-uniform rank allocation strategies, and provide a lightweight proxy for domain shift by varying calibration data. Figure 9 compares base and unlearned models using weighted average explained variance (EV) and the energy-weighted CDF of effective ranks. Across both Qwen 0.6B and LLaMA 3.1 8B, the unlearned models consistently achieve higher EV at smaller ranks and concentrate more representational energy in low-rank subspaces. In contrast, the base models require larger ranks to capture the same variance. The effect is modest in Qwen 0.6B but pronounced in LLaMA 3.1 8B, where the unlearned variant is markedly more low-rank efficient. Overall, unlearning improves LoRA efficiency, enabling comparable adaptation with fewer parameters.

### A.8 VARYING RETAIN DATASET SIZE WITH VARYING $\sigma$

To assess the effect of forget-set size, we varied it across 1k, 3k, 5k, 7k, and 9k examples with varying values of $\sigma$ 0, 0.5, 1.0. The forget set was BookCorpus and the eval/retain datset is PubMedQA and MedMCQA. Experiments were conducted with Qwen-0.6B and LlaMA 3.1-8B to test whether the effects are consistent across models of different sizes and architectures.

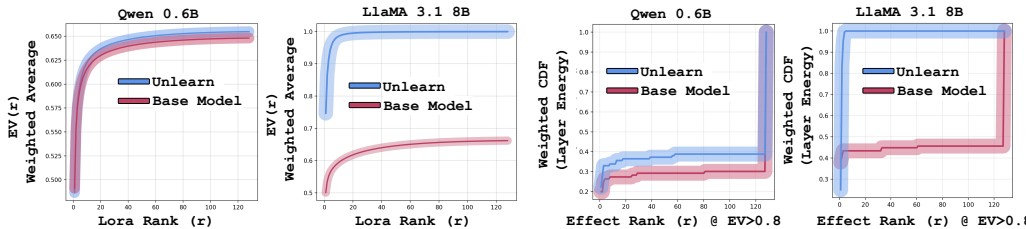

Figure 9: Spectral surrogate analysis for LoRA capacity. We compute explained variance curves from activations and gradients, aggregate across layers, and extract effective rank estimates. Unlearned models concentrate more energy in low-rank subspaces, making them more LoRA-efficient than their base counterparts.

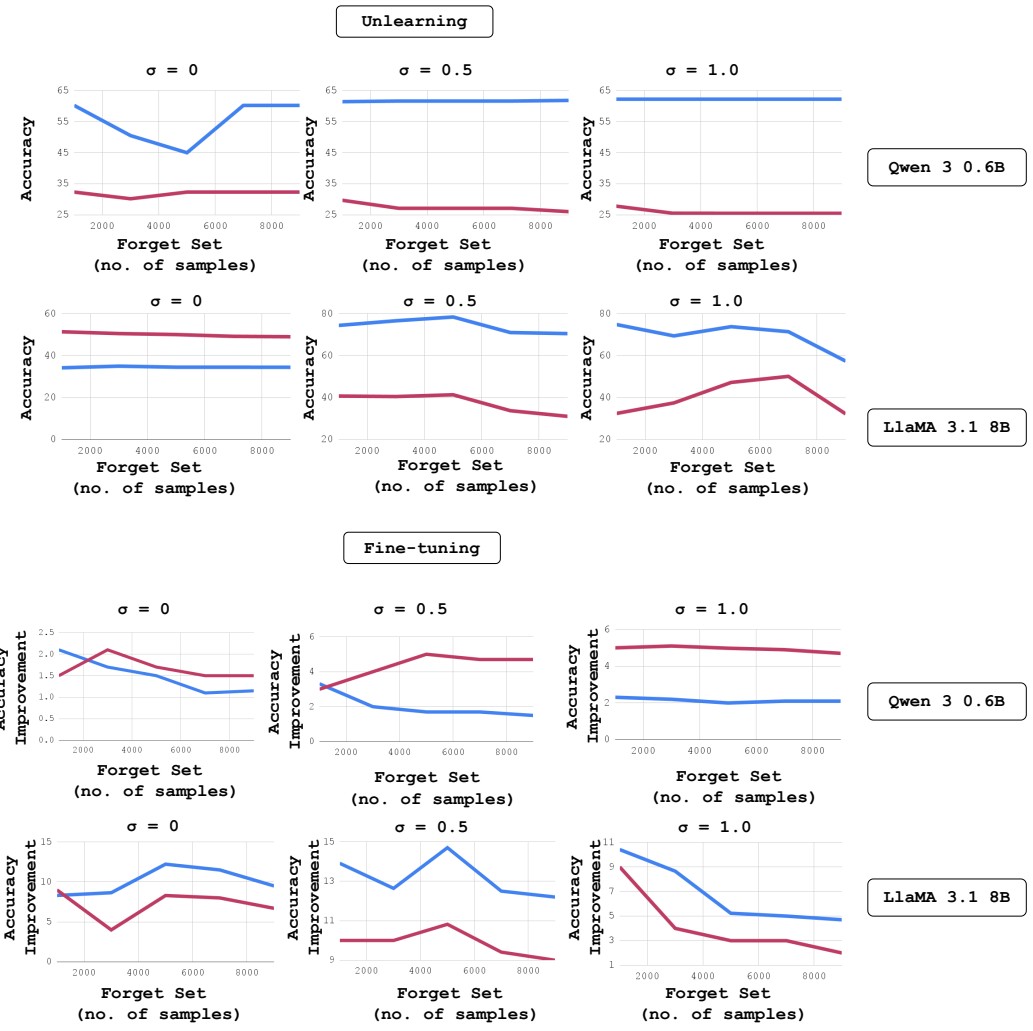

Figure 10: Effect of varying retain dataset size with varying $\sigma$ on accuracy after unlearning and accuracy improvement after fine-tuning.s

For Qwen-0.6B, increasing $\sigma$ from 0 to 1.0 stabilized retention on both datasets: PubMedQA remained consistently high ( 60–65%) across all forget-set sizes, while MedMCQA showed a smaller decline, indicating that $\sigma$ noise helps prevent catastrophic forgetting. For LLaMA-3.1-8B, retention was best maintained at moderate $\sigma$ ($\sigma$=0.5), with PubMedQA accuracy remaining above 65% and

MedMCQA showing stable trends, whereas high $\sigma$ ($\sigma$=1.0) led to retention collapse as forget-set size increased.

Fine-tuning further emphasized these architecture-dependent differences: Qwen-0.6B exhibited modest but stable improvements under noise, while LLaMA-3.1-8B achieved larger gains at $\sigma$=0.5 but deteriorated sharply at $\sigma$=1.0. These results demonstrate that while noise injection facilitates stable unlearning, retention behavior is dataset- and model-dependent, with smaller models benefiting from stronger $\sigma$ and larger models requiring moderate $\sigma$ to preserve generalization. We find a clear scaling effect: larger models require larger forget sets to achieve comparable unlearning, reflecting their greater capacity to store diverse knowledge. In contrast, smaller models can be effectively unlearned with much smaller sets.

### A.9 EFFECT OF DATASET SIZE AND GA WEIGHT

Accuracy improvement on PubMedQA (blue) and MedMCQA (red) is shown as a function of the retain set size under two $\lambda$ values. Both models achieve substantial gains at $\lambda$=0.5, with consistent improvements as the retain set grows. In contrast, $\lambda$=1.0 severely limits improvement, particularly for larger retain sets, indicating that moderate GA weighting better balances knowledge retention and adaptation across datasets and architectures.

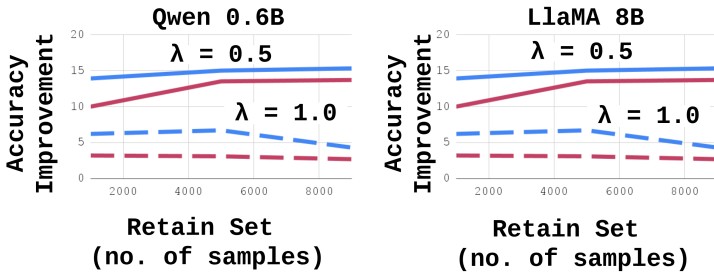

Figure 11: Effect of dataset size and gradient ascent weight,$\lambda$ on accuracy improvement

### A.10 GA WEIGHT AND GD WEIGHT

Accuracy improvement increases with $\lambda$ up to 0.5 for both datasets, after which gains taper off, suggesting diminishing returns from higher weighting. Right: Accuracy improvement remains relatively stable across $\sigma$ values, indicating that GD weighting exerts weaker influence on fine-tuning gains compared to GA. These results suggest that GA weighting ($\lambda$) plays a more critical role than GD weighting ($\sigma$) in mediating retention gains during fine-tuning, with $\lambda$=0.5 emerging as the optimal trade-off point.

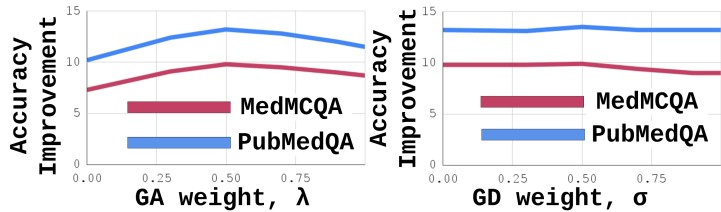

Figure 12: Effect of gradient descent weight,$\sigma$ and gradient ascent weight,$\lambda$ on accuracy improvement.

## B EFFECT OF LEARNING RATE ON UNLEARNING AND FINE-TUNING OF F2F

Table 8 shows the sensitivity analysis of unlearning and finetuning on th performance. For unlearning, performance on PubMedQA and MedMCQA increases as we raise the learning rate from

$5 \times 10^{-6}$ to $3 \times 10^{-5}$. PubMedQA accuracy improves by +5.30 points, while MedMCQA increases by +12.48 points. We chose $3 \times 10^{-5}$ as there was no performance gain after this.

For fine-tuning as we increase the learning rate from $5 \times 10^{-6}$ to $2 \times 10^{-5}$, accuracy on Pub-MedQA improves by +5.70 points, and MedMCQA increases by +7.25 points. Beyond $2 \times 10^{-5}$, larger learning rates do not bring further improvements. Hence, these results motivate the choice of learning rate of $3 \times 10^{-5}$ for unlearning and $2 \times 10^{-5}$ for fine-tuning.

Table 8: Learning rate ablation for Unlearning and Fine-Tuning on LLaMA 3.1 8B-Instruct.

| Unlearning | | | Fine-Tuning | | |
|---|---|---|---|---|---|
| Learning Rate | PubMedQA | MedMCQA | Learning Rate | PubMedQA | MedMCQA |
| $5 \times 10^{-6}$ | 73.91 | 48.02 | $5 \times 10^{-6}$ | 84.20 | 63.00 |
| $1 \times 10^{-5}$ | 75.20 | 51.22 | $1 \times 10^{-5}$ | 85.22 | 65.05 |
| $1.5 \times 10^{-5}$ | 76.99 | 54.81 | $1.5 \times 10^{-5}$ | 86.71 | 67.82 |
| $2 \times 10^{-5}$ | 76.81 | 58.35 | $2 \times 10^{-5}$ | 89.90 | 70.25 |
| $2.5 \times 10^{-5}$ | 79.05 | 59.21 | $2.5 \times 10^{-5}$ | 89.85 | 67.09 |
| $3 \times 10^{-5}$ | 79.21 | 60.50 | $3 \times 10^{-5}$ | 89.75 | 69.91 |
| $4 \times 10^{-5}$ | 78.00 | 60.33 | $4 \times 10^{-5}$ | 88.00 | 69.15 |

## C  EFFECT WITH MULTI-SEED SETTINGS

Table 9 demonstrates that the F2F consistently improves over SFT across models, domains, and benchmarks under multi-seed evaluation. In the coding domain, F2F adds between +1.8 and +9.4 pass@1 points on top of SFT for both Qwen 2.5 7B and LLaMA-3.1 8B, despite SFT already providing substantial gains over the Base models.

In the medical domain, F2F similarly yields consistent improvements, with gains of +3.5 to +6.0 accuracy points over SFT on PubMedQA and MedMCQA. Across all configurations, the standard deviations are very small, indicating that F2F's advantages are robust to random seed variation rather than arising from unstable or lucky runs.

Table 9: Multi-seed robustness of F2F. We report mean $\pm$ std over 3 seeds. Base is single-seed (no SFT). SFT and F2F are averaged across 3 seeds under identical settings.

| Model | Domain | Metric | Base | SFT (3 seeds) | F2F (3 seeds) | $\Delta$ (F2F–SFT) |
|---|---|---|---|---|---|---|
| **Coding** | | | | | | |
| Qwen 2.5 7B | Coding | HumanEval pass@1 (%) | 40.20 | $44.35 \pm 0.004$ | $53.70 \pm 0.005$ | +9.4 |
| Qwen 2.5 7B | Coding | MBPP pass@1 (%) | 65.85 | $72.90 \pm 0.007$ | $76.25 \pm 0.004$ | +3.4 |
| LLaMA-3.1 8B | Coding | HumanEval pass@1 (%) | 33.54 | $57.37 \pm 0.004$ | $61.59 \pm 0.005$ | +4.2 |
| LLaMA-3.1 8B | Coding | MBPP pass@1 (%) | 49.00 | $58.95 \pm 0.005$ | $60.73 \pm 0.003$ | +1.8 |
| **Medical** | | | | | | |
| Qwen 2.5 7B | Medical | PubMedQA acc. (%) | 73.00 | $81.52 \pm 0.012$ | $85.00 \pm 0.007$ | +3.5 |
| Qwen 2.5 7B | Medical | MedMCQA acc. (%) | 56.23 | $65.35 \pm 0.008$ | $69.35 \pm 0.002$ | +4.0 |
| LLaMA-3.1 8B | Medical | PubMedQA acc. (%) | 75.20 | $89.51 \pm 0.006$ | $91.35 \pm 0.009$ | +1.8 |
| LLaMA-3.1 8B | Medical | MedMCQA acc. (%) | 59.43 | $64.55 \pm 0.001$ | $70.55 \pm 0.001$ | +6.0 |

### C.1  EFFECT OF UNLEARNING STEP SIZE ON COMPUTATIONAL COST AND ACCURACY

Figure 13 highlights the change in medical domain benchmarks' accuracy with varying unlearning step counts and shows how runtime also changes with it. Figure shows that as the number of unlearning steps increases, accuracy on the medical domain benchmarks remains essentially unchanged, while runtime grows. Accordingly, we choose 1,000 unlearning steps as a practical default: it matches the best accuracy yet requires only 0.55 GPU-hours on a single A100 GPU.

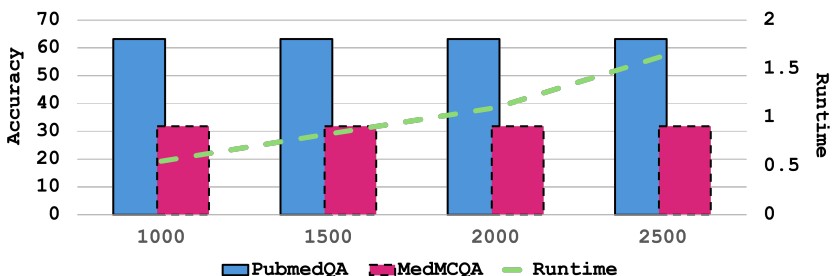

Figure 13: Runtime across different unlearning step counts, measured on a single A100 in GPU-hours.

