# OpenReview forum: "Forget-to-Focus: Can unlearning Improve Domain Specialization in LLMs?"
_ICLR.cc/2026/Conference — Submitted to ICLR 2026_

### Official Review · Reviewer_P5Ts · 2025-10-31

**Soundness:** 2
**Presentation:** 3
**Contribution:** 3
**Rating:** 4
**Confidence:** 3

**Summary:**

This paper introduces "Forget-to-Focus," a novel approach that first employs unlearning methods and then fine-tunes LLMs on specific domains, outperforming traditional SFT. The authors support their method with both a mathematical proof and a latent space analysis.

**Strengths:**

（1）The story, and motivation is well presented. The writing is easy to follow.
（2）The findings are meaningful, which shapes unlearning as a way to improve domain specification.
（3）The experiments cover models with different sizes, architectures, and different domains.

**Weaknesses:**

（1）The choice of base model seems weird and needs justification. Firstly, the chosen model includes different versions (e.g. Qwen2 72B Instruct vs Qwen3-0.6B, Llama2-13B and Llama3.1-8B-Instruct). Can the authors try Qwen2.5 series with different model sizes? Moreover, Qwen3-0.6B is not an instruct model.
（2）Why in Table1, finetuned Gemma-2B-Instruct is worse then the base model? All the finetuning methods, namely SFT, DAPT, LORA, CurlLora, makes the performance worse.  I doubt the correctness of finetuning process, or the hyperparam choice.
（3）The authors do not give a automated method to choose forget set, which limits the value of proposed method. For example, given a new domain, how to choose the forget set?

**Questions:**

(1)	Table1 is confusing. The authors should use more clear notations other than label of (1) and (2). Same problem exists for Table3.
(2)	In Table2, is it Llama3.1-8B-Instruct instead of “LLaMA 8B”? (also for L288, L327, L351, etc.)

---

> ### Author Response · Authors · 2025-11-18
> **Response to Reviewer - P5Ts [1]**
>
> We thank reviewer P5Ts for constructive review. We appreciate your time and feedback to enhance our paper’s quality. We address the reviewer’s concerns below. Updated texts are in **blue** in the main paper.
>
> **W-1 : “Base model choices (Qwen versions, mixed instruct/base).”**
>
> Thank you for pointing out the diversity of base models. Our intent was to demonstrate **family- and scale-robustness**. For that reason, we intentionally covered three major families (Qwen, Llama, Gemma) and a wide range of sizes (from hundreds of millions to tens of billions of parameters). However, we agree that Qwen2.5 would be an interesting additional testbed.Hence, we have: (a) added **Qwen-2.5** mid-scale variants in the supplementary, and (b) made the motivation explicit in §3.3.
>
> **MBPP and HumanEval pass@1 across Qwen 2.5 Instruct variants and tuning methods**
>
> | Coding Method | Qwen 2.5 1.5B MBPP | Qwen 2.5 1.5B HumanEval | Qwen 2.5 7B MBPP | Qwen 2.5 7B HumanEval | Qwen 2.5 14B MBPP | Qwen 2.5 14B HumanEval |
> | --- | --- | --- | --- | --- | --- | --- |
> | (1) Base Model | 40.00 | 35.37 | 65.85 | 40.20 | 62.20 | 36.80 |
> | (1) + SFT | 45.04 | 38.25 | 72.53 | 43.80 | 65.75 | 40.55 |
> | (1) + DAPT | 46.00 | 41.03 | 71.35 | 44.65 | 69.00 | 41.69 |
> | (1) + LORA | 44.76 | 39.01 | 70.33 | 44.83 | 67.35 | 49.97 |
> | (1) + CurlLora | 46.22 | 43.21 | 72.00 | 45.09 | 68.00 | 41.33 |
> | (2) Unl_{GA+GD} | 43.00 | 37.11 | 65.60 | 45.10 | 64.55 | 40.25 |
> | (2) + SFT | 53.80 | 47.30 | 75.90 | 53.20 | 71.30 | 51.59 |
> | (3) Unl_{GA} | 39.61 | 34.30 | 67.02 | 43.21 | 61.11 | 37.81 |
> | (3) + SFT | 52.40 | 45.80 | 72.50 | 48.70 | 68.45 | 48.10 |
>
> **W-2 : Gemma-2B-Instruct finetuning underperforms the base model in Table 1**
>
> Table 1 does *not* show that all finetuning variants hurt Gemma.**F2F+SFT** *improves* Gemma-2B over the base on both benchmarks: the base model scores **19.80 / 16.46** (MBPP / HumanEval), while F2F+SFT reaches **20.05 / 21.30**, which is the **best** result among all methods for Gemma and is highlighted accordingly in the table. Similarly, (1)+DAPT and CurlLoRA improve at least one of the two metrics. The rows with strong degradation for Gemma (e.g., “GA+GD” alone) correspond to the only **unlearning checkpoints in our F2F method.** We clarified clarify this more explicitly in the text and caption.
>
> **W-3 : Automated forget-set construction for new domains**
> Thank you for pointing this out. In revision, we provide this automated method. [text in **red** in updated paper]
>
> For this we compute the embedding of the target domain and the broader samples, and we can find the similarity using Cosine distance to get the target domain samples.
>
> We encode each data point $x$ with a Transformer :
>
> $$
> h_x = f_{\theta}(x),
> $$
>
> and define the target-domain centroid as :
>
> $$
> c_T = \frac{1}{|D_T|} \sum_{x' \in D_T} f_{\theta}(x'),
> $$
>
> then rank samples by the cosine distance : (higher distance means lower similarity)
>
> $$
> d_{\cos}(x) = 1 - \frac{h_x^\top c_T}{\lVert h_x \rVert \, \lVert c_T \rVert}.
> $$
>
> We have evaluated this and added the evaluation result in the below table (BC-Cosine is newly added forget set selection method) and in **Table 3** in main text.
>
> **Model: Qwen3 0.6B**
>
> | FD | Method | MBPP | HumanEval | PubMedQA | MedMCQA | Hendrycks-MATH | GSM8K |
> | --- | --- | --- | --- | --- | --- | --- | --- |
> | BC-Select | (1) GA+GD | 30.00 | 21.34 | 61.60 | 31.44 | 39.07 | 0.26 |
> | BC-Select | (1) + Tuning | 31.60 | 42.07 | 69.60 | 45.31 | 54.11 | 15.30 |
> | BC-Select | (2) GA | 24.00 | 20.73 | 60.40 | 31.29 | 25.09 | 0.24 |
> | BC-Select | (2) + SFT | 31.60 | 40.02 | 58.80 | 40.26 | 51.20 | 14.00 |
> | BC-Mixed | (1) GA+GD | 24.20 | 20.12 | 61.80 | 30.38 | 31.47 | 0.06 |
> | BC-Mixed | (1) + Tuning | 29.90 | 40.00 | 60.20 | 23.31 | 52.00 | 13.21 |
> | BC-Mixed | (2) GA | 23.80 | 20.12 | 60.20 | 31.89 | 25.00 | 0.00 |
> | BC-Mixed | (2) + Tuning | 28.00 | 33.10 | 61.20 | 35.43 | 50.00 | 13.51 |
> | BC-Cosine | (1) UA+GD | 24.01 | 18.00 | 61.20 | 29.32 | 29.05 | 0.10 |
> | BC-Cosine | (1) + Tuning | 31.55 | 41.90 | 67.33 | 45.00 | 54.85 | 13.20 |
> | BC-Cosine | (2) GA | 22.22 | 18.23 | 58.45 | 29.00 | 25.00 | 0.00 |
> | BC-Cosine | (2) + Tuning | 29.91 | 40.50 | 57.65 | 38.75 | 48.90 | 13.07 |
> | - | (3) Baseline | 22.60 | 19.50 | 57.80 | 32.25 | 41.09 | 0.02 |
> | - | (3) + Tuning | 28.80 | 31.71 | 62.60 | 42.12 | 50.06 | 12.30 |

---

> ### Author Response · Authors · 2025-11-18
> **Response to Reviewer - P5Ts [2]**
>
> **Model: LLaMA3.1 8B**
>
> | FD | Method | MBPP | HumanEval | PubMedQA | MedMCQA | Hendrycks-MATH | GSM8K |
> | --- | --- | --- | --- | --- | --- | --- | --- |
> | BC-Select | (1) GA+GD | 43.60 | 54.88 | 79.21 | 60.50 | 18.00 | 61.70 |
> | BC-Select | (1) + Tuning | 60.10 | 60.37 | 89.90 | 70.25 | 29.50 | 70.51 |
> | BC-Select | (2) GA | 22.60 | 1.20 | 75.22 | 60.20 | 10.51 | 50.91 |
> | BC-Select | (2) + Tuning | 58.66 | 57.70 | 87.00 | 67.15 | 25.70 | 67.20 |
> | BC-Mixed | (1) GA+GD | 40.50 | 52.52 | 79.50 | 59.30 | 17.00 | 51.00 |
> | BC-Mixed | (1) + Tuning | 56.20 | 55.76 | 87.61 | 70.10 | 28.81 | 65.20 |
> | BC-Mixed | (2) GA | 33.20 | 25.50 | 72.30 | 57.00 | 5.20 | 35.20 |
> | BC-Mixed | (2) + Tuning | 52.30 | 40.90 | 86.90 | 61.20 | 23.01 | 66.15 |
> | BC-Cosine | (1) GA+GD | 42.55 | 53.76 | 79.00 | 58.22 | 17.91 | 61.00 |
> | BC-Cosine | (1) + Tuning | 59.55 | 59.86 | 85.31 | 71.02 | 28.33 | 68.57 |
> | BC-Cosine | (2) GA | 20.35 | 0.90 | 73.05 | 58.99 | 9.32 | 50.00 |
> | BC-Cosine | (2) + Tuning | 57.76 | 57.55 | 85.31 | 66.00 | 27.01 | 67.10 |
> | - | (3) Baseline | 49.00 | 33.54 | 75.20 | 59.43 | 19.90 | 62.85 |
> | - | (3) + Tuning | 56.60 | 56.71 | 85.31 | 64.20 | 25.51 | 66.70 |
>
> **Model: LLaMA2 13B**
>
> | FD | Method | MBPP | HumanEval | PubMedQA | MedMCQA | Hendrycks-MATH | GSM8K |
> | --- | --- | --- | --- | --- | --- | --- | --- |
> | BC-Select | (1) GA+GD | 27.22 | 0.60 | 74.90 | 38.68 | 29.00 | 5.10 |
> | BC-Select | (1) + Tuning | 50.31 | 46.15 | 90.11 | 60.10 | 51.50 | 21.50 |
> | BC-Select | (2) GA | 0.00 | 25.50 | 70.00 | 36.51 | 24.35 | 2.00 |
> | BC-Select | (2) + Tuning | 45.01 | 44.70 | 89.33 | 57.93 | 50.90 | 20.00 |
> | BC-Mixed | (1) GA+GD | 27.20 | 0.45 | 73.00 | 37.50 | 27.00 | 5.10 |
> | BC-Mixed | (1) + Tuning | 47.50 | 45.91 | 89.55 | 61.30 | 50.30 | 20.00 |
> | BC-Mixed | (2) GA | 0.00 | 10.00 | 65.99 | 29.55 | 23.55 | 1.05 |
> | BC-Mixed | (2) + Tuning | 39.55 | 40.01 | 87.00 | 50.60 | 47.60 | 20.00 |
> | BC-Cosine | (1) GA+GD | 25.30 | 0.52 | 73.44 | 37.62 | 29.09 | 6.30 |
> | BC-Cosine | (1) + Tuning | 48.91 | 43.30 | 89.00 | 58.42 | 50.33 | 18.03 |
> | BC-Cosine | (2) GA | 0.00 | 23.33 | 69.55 | 36.22 | 24.00 | 0.92 |
> | BC-Cosine | (2) + Tuning | 44.33 | 42.05 | 89.10 | 60.99 | 49.62 | 19.52 |
> | - | (3) Baseline | 27.22 | 0.60 | 75.20 | 38.68 | 27.50 | 2.00 |
> | - | (3) + Tuning | 37.01 | 40.21 | 86.30 | 54.00 | 37.09 | 16.30 |
>
> **Q-1**  We have updated this in our updated manuscript.

---

> ### Author Response · Authors · 2025-11-27
>
> Dear Reviewer P5Ts,
>
> We hope you’re doing well. We wanted to kindly check if there are any updates or if additional clarification or update are needed from our side. We understand everyone’s busy, but we’d really appreciate any feedback or guidance on the current status of our rebuttal.
>
> Thank you very much for your time and effort reviewing our work.

---

> > ### Comment · Reviewer_P5Ts · 2025-11-27
> >
> > Thanks for the responses from the authors, which addressed most of my concerns. I'll keep my score unchanged.

---

### Official Review · Reviewer_LotK · 2025-10-31

**Soundness:** 3
**Presentation:** 3
**Contribution:** 3
**Rating:** 6
**Confidence:** 3

**Summary:**

Forget-to-Focus (F2F) proposes a two-stage fine-tuning protocol where a large language model (LLM) first undergoes targeted unlearning of irrelevant general-domain knowledge before domain-specific fine-tuning. The approach aims to mitigate negative transfer from pretraining priors that hinder specialization, reframing unlearning as a capacity reallocation mechanism rather than a privacy safeguard. Formally, F2F minimizes an objective combining gradient ascent on a forget set and gradient descent on a retain set, producing a new initialization that lies closer to the downstream optimum. Theoretical analysis shows contraction of irrelevant components in parameter space, while empirical results demonstrate consistent improvements across coding, mathematical, and medical domains. Representational analyses (CKA, SVCCA, Fisher information) confirm that unlearning reshapes the model’s geometry toward domain-relevant subspaces and improves calibration.

**Strengths:**

- The paper introduces a conceptually novel perspective—using unlearning not for data privacy, but as a preparatory step for specialization. This reframing is both original and intuitively appealing, addressing a key limitation of conventional fine-tuning.
- Provides a mathematical formulation of how unlearning alters optimization dynamics, supported by a contraction analysis that theoretically guarantees reduced distance to domain optima.
- Empirical validation is comprehensive: spanning multiple LLM families (Qwen, LLaMA, Gemma) and domains (medical, math, coding) up to 72B parameters, ensuring robustness of findings. Also, authors incorporates representational diagnostics (CKA, SVCCA, Fisher analysis, PCA-shift) that offer interpretable empirical evidence of capacity redistribution and geometry realignment after unlearning.
- Quantitative performance gains—up to +32.5% in coding (HumanEval) and +11.9% on large models (Qwen-72B)—as well as improved calibration and reduced overconfidence on sensitive QA tasks.

**Weaknesses:**

1. **Dependence on forget/retain set quality.**
Performance improvements are sensitive to how cleanly the forget set separates general and domain-relevant data. The method assumes such partitioning is feasible, but in real-world setups, this boundary is often ambiguous. The paper lacks guidelines for constructing forget sets in open-domain contexts.
2. **Simplistic convex surrogate assumption.**
The theoretical guarantees rely on convexity and orthogonal subspace decomposition, which may not hold in deep nonlinear LLMs. While insightful, the analysis may not fully capture realistic optimization dynamics.
3. **Limited diversity of domain evaluation.**
Despite testing on three domains, most benchmarks are standard (PubMedQA, MATH, HumanEval) and measure only accuracy. The paper does not probe harder or more diverse domain shifts (e.g., adversarial or multimodal settings), which limits generalizability claims.
4. **Computational trade-offs unquantified.**
Although the authors mention unlearning adds modest cost (~1k steps), the paper does not analyze scaling behavior or potential instability when applied to trillion-parameter LLMs or multi-domain settings.

**Questions:**

1. How robust is the proposed benefit of unlearning to imperfect or noisy forget/retain set selection? Could overlap between general and domain-relevant data negate the observed gains?

2. The theoretical analysis relies on convex surrogate assumptions and orthogonal subspace decomposition. How well do these simplifications capture the optimization dynamics of non-convex LLM training?

3. Could the unlearning–retuning pipeline be extended to multi-domain or continual adaptation, where domains overlap or evolve over time, without catastrophic interference?

---

> ### Author Response · Authors · 2025-11-18
> **Response to Reviewer - LotK [1]**
>
> We thank reviewer LotK for taking their valuable time to review our paper and give feedback while highlighting the strengths. We address each points below :
>
> **W-1 : “Dependence on forget/retain quality.”**
>
> F2F does **not** require a clean or binary separation; in our experiments we already operate with coarse, noisy splits (Bc-Mixed and BC-Select , BC-Cosine (newly added), yet we still see consistent gains across model families and sizes.
>
> We now provide automated curation of forget set and for this we compute the embedding of the target domain and the broader samples, and we can find the similarity using Cosine distance to get the target domain samples.
>
> We encode each data point $x$ with a Transformer :
>
> $$
> h_x = f_{\theta}(x),
> $$
>
> and define the target-domain centroid as :
>
> $$
> c_T = \frac{1}{|D_T|} \sum_{x' \in D_T} f_{\theta}(x'),
> $$
>
> then rank samples by the cosine distance : (higher distance means lower similarity)
>
> $$
> d_{\cos}(x) = 1 - \frac{h_x^\top c_T}{\lVert h_x \rVert \, \lVert c_T \rVert}.
> $$
>
> We have evaluated this and added the evaluation result in the below table (BC-Cosine is newly added forget set selection method) and in **Table 3** in main text.
>
> **Model: Qwen3 0.6B**
>
> | FD | Method | MBPP | HumanEval | PubMedQA | MedMCQA | Hendrycks-MATH | GSM8K |
> | --- | --- | --- | --- | --- | --- | --- | --- |
> | BC-Select | (1) GA+GD | 30.00 | 21.34 | 61.60 | 31.44 | 39.07 | 0.26 |
> | BC-Select | (1) + Tuning | 31.60 | 42.07 | 69.60 | 45.31 | 54.11 | 15.30 |
> | BC-Select | (2) GA | 24.00 | 20.73 | 60.40 | 31.29 | 25.09 | 0.24 |
> | BC-Select | (2) + SFT | 31.60 | 40.02 | 58.80 | 40.26 | 51.20 | 14.00 |
> | BC-Mixed | (1) GA+GD | 24.20 | 20.12 | 61.80 | 30.38 | 31.47 | 0.06 |
> | BC-Mixed | (1) + Tuning | 29.90 | 40.00 | 60.20 | 23.31 | 52.00 | 13.21 |
> | BC-Mixed | (2) GA | 23.80 | 20.12 | 60.20 | 31.89 | 25.00 | 0.00 |
> | BC-Mixed | (2) + Tuning | 28.00 | 33.10 | 61.20 | 35.43 | 50.00 | 13.51 |
> | BC-Cosine | (1) UA+GD | 24.01 | 18.00 | 61.20 | 29.32 | 29.05 | 0.10 |
> | BC-Cosine | (1) + Tuning | 31.55 | 41.90 | 67.33 | 45.00 | 54.85 | 13.20 |
> | BC-Cosine | (2) GA | 22.22 | 18.23 | 58.45 | 29.00 | 25.00 | 0.00 |
> | BC-Cosine | (2) + Tuning | 29.91 | 40.50 | 57.65 | 38.75 | 48.90 | 13.07 |
> | - | (3) Baseline | 22.60 | 19.50 | 57.80 | 32.25 | 41.09 | 0.02 |
> | - | (3) + Tuning | 28.80 | 31.71 | 62.60 | 42.12 | 50.06 | 12.30 |
>
> **Model: LLaMA3.1 8B**
>
> | FD | Method | MBPP | HumanEval | PubMedQA | MedMCQA | Hendrycks-MATH | GSM8K |
> | --- | --- | --- | --- | --- | --- | --- | --- |
> | BC-Select | (1) GA+GD | 43.60 | 54.88 | 79.21 | 60.50 | 18.00 | 61.70 |
> | BC-Select | (1) + Tuning | 60.10 | 60.37 | 89.90 | 70.25 | 29.50 | 70.51 |
> | BC-Select | (2) GA | 22.60 | 1.20 | 75.22 | 60.20 | 10.51 | 50.91 |
> | BC-Select | (2) + Tuning | 58.66 | 57.70 | 87.00 | 67.15 | 25.70 | 67.20 |
> | BC-Mixed | (1) GA+GD | 40.50 | 52.52 | 79.50 | 59.30 | 17.00 | 51.00 |
> | BC-Mixed | (1) + Tuning | 56.20 | 55.76 | 87.61 | 70.10 | 28.81 | 65.20 |
> | BC-Mixed | (2) GA | 33.20 | 25.50 | 72.30 | 57.00 | 5.20 | 35.20 |
> | BC-Mixed | (2) + Tuning | 52.30 | 40.90 | 86.90 | 61.20 | 23.01 | 66.15 |
> | BC-Cosine | (1) GA+GD | 42.55 | 53.76 | 79.00 | 58.22 | 17.91 | 61.00 |
> | BC-Cosine | (1) + Tuning | 59.55 | 59.86 | 85.31 | 71.02 | 28.33 | 68.57 |
> | BC-Cosine | (2) GA | 20.35 | 0.90 | 73.05 | 58.99 | 9.32 | 50.00 |
> | BC-Cosine | (2) + Tuning | 57.76 | 57.55 | 85.31 | 66.00 | 27.01 | 67.10 |
> | - | (3) Baseline | 49.00 | 33.54 | 75.20 | 59.43 | 19.90 | 62.85 |
> | - | (3) + Tuning | 56.60 | 56.71 | 85.31 | 64.20 | 25.51 | 66.70 |
>
> **Model: LLaMA2 13B**
>
> | FD | Method | MBPP | HumanEval | PubMedQA | MedMCQA | Hendrycks-MATH | GSM8K |
> | --- | --- | --- | --- | --- | --- | --- | --- |
> | BC-Select | (1) GA+GD | 27.22 | 0.60 | 74.90 | 38.68 | 29.00 | 5.10 |
> | BC-Select | (1) + Tuning | 50.31 | 46.15 | 90.11 | 60.10 | 51.50 | 21.50 |
> | BC-Select | (2) GA | 0.00 | 25.50 | 70.00 | 36.51 | 24.35 | 2.00 |
> | BC-Select | (2) + Tuning | 45.01 | 44.70 | 89.33 | 57.93 | 50.90 | 20.00 |
> | BC-Mixed | (1) GA+GD | 27.20 | 0.45 | 73.00 | 37.50 | 27.00 | 5.10 |
> | BC-Mixed | (1) + Tuning | 47.50 | 45.91 | 89.55 | 61.30 | 50.30 | 20.00 |
> | BC-Mixed | (2) GA | 0.00 | 10.00 | 65.99 | 29.55 | 23.55 | 1.05 |
> | BC-Mixed | (2) + Tuning | 39.55 | 40.01 | 87.00 | 50.60 | 47.60 | 20.00 |
> | BC-Cosine | (1) GA+GD | 25.30 | 0.52 | 73.44 | 37.62 | 29.09 | 6.30 |
> | BC-Cosine | (1) + Tuning | 48.91 | 43.30 | 89.00 | 58.42 | 50.33 | 18.03 |
> | BC-Cosine | (2) GA | 0.00 | 23.33 | 69.55 | 36.22 | 24.00 | 0.92 |
> | BC-Cosine | (2) + Tuning | 44.33 | 42.05 | 89.10 | 60.99 | 49.62 | 19.52 |
> | - | (3) Baseline | 27.22 | 0.60 | 75.20 | 38.68 | 27.50 | 2.00 |
> | - | (3) + Tuning | 37.01 | 40.21 | 86.30 | 54.00 | 37.09 | 16.30 |

---

> ### Author Response · Authors · 2025-11-18
> **Response to Reviewer - LotK [2]**
>
> **W-2 and Q-2: “Theory uses convex surrogate / orthogonal subspaces.”**
>
> The use of orthogonal gradient/subspace decompositions to model inference between tasks has been widely used and adopted in continual learning and LLM adaptation. In the paper, the analysis is meant to provide qualitative guidance: (i) when the forget set induces gradients that systematically oppose the domain-preferred direction, and (ii) how subtracting that component reallocates capacity in representation space.
> We pair the analysis with empirical evidence: we show (in main text and supplementary) representation-space shifts, consistent performance trends across multiple architectures and domains, and ablations over GA/GD weights that match the theoretical predictions. Our current work focuses on text-only LLMs so, multimodal settings are out of scope for this work.
>
> **W-3 : “Limited diversity of domains.”**
>
> Our choice of **code** (MBPP/HumanEval), **biomedical QA** (PubMedQA/MedMCQA), and **math** (GSM8K/MATH) was deliberate: these are **widely-used**, **challenging benchmarks** that capture **qualitatively different** skills (program synthesis, factual scientific reasoning, multi-step symbolic reasoning) and are standard proxies for domain-specialized capabilities. Within each domain we include multiple datasets and multiple model families/sizes to test robustness of the effect, rather than optimizing for a single bespoke shift.
>
> **We will appreciate any suggested domains or benchamark datasets to include in our work**
>
> **Q-1 : How robust is the benefit of unlearning to imperfect or noisy forget/retain set selection? Could overlap between general and domain-relevant data negate the gains?**
>
> Our method does *not* require a perfectly clean partition; it only requires that, on average, the forget set be *less aligned* with the target domain than the retain set. In practice, our experiments already operate with **noisy splits** BC-Select vs. BC-mixed vs. BC-Cosine (newly added), yet we observe consistent improvements across model families and sizes.
>
> **Q-3 : Can the unlearning–retuning pipeline extend to multi-domain or continual adaptation with overlapping/evolving domains, without catastrophic interference?**
>
> Conceptually, yes: F2F naturally extends to multi-domain or continual settings by treating each domain as a slice of the data mixture and applying the same **gradient-alignment principle**. In a multi-domain scenario, one can (i) identify, for each target domain, which portions of the mixture are antagonistic (negative alignment) versus supportive (positive alignment), and (ii) run F2F step that down-weights or forgets only the *most conflicting* components while preserving shared, useful structure. While this is out of scope of our paper, **we will include it as a promising direction for future work.**

---

> > ### Comment · Reviewer_LotK · 2025-11-25
> >
> > Thank you for the detailed responses to my questions. I appreciate the clarifications and the additional analyses provided. While the authors have addressed the raised concerns, I would like to emphasize that the weaknesses I pointed out are not fatal issues but they do meaningfully qualify the scope and generality of the contributions. Given this, and considering the overall novelty, empirical coverage, and modeling assumptions, I believe my current score already reflects a fair and balanced assessment of the work. Therefore, I will keep my original rating.

---

> > > ### Author Response · Authors · 2025-11-26
> > > **Response to Reviewer - LotK**
> > >
> > > Thank you so much for your kind response. You comments have helped us enhance the quality of the paper.

---

### Official Review · Reviewer_N1qK · 2025-11-01

**Soundness:** 2
**Presentation:** 3
**Contribution:** 2
**Rating:** 4
**Confidence:** 4

**Summary:**

The paper proposes Forget-to-Focus (F2F): first “unlearn” with a forget set (and small retain set), then fine-tune on the target domain. They report gains on coding (MBPP/HumanEval), medical QA (PubMedQA/MedMCQA), and math (GSM8K/MATH), and show representation-level shifts after unlearning.

**Strengths:**

It is an interesting idea and the paper presents clear, simple two-stage protocol and motivation around negative transfer. The paper try to evaluate broad set of domains and model sizes.

**Weaknesses:**

- Feasibility of “forgetting general data.”
The paper itself admits it’s hard to decide what to forget without hurting useful priors.


- Risk to general capabilities/communication.
They only audit multiple-choice commonsense tasks, not instruction-following or dialog quality. So we can’t tell how chatty/instructional behavior survives F2F—even though the base models are “Instruct” variants. The appendix “broad skills” audit doesn’t answer that.
Instruction following is unreported, such as  MT-Bench / AlpacaEval / Arena-Hard-style evaluation.

- SFT sensitivity / insufficient controls.
F2F’s headline gains could be sensitive to SFT choices (epochs, LR, batch, data subsampling). The paper provides a single-recipe config and some ablations (GA/GD weights, retain size), but no multi-seed variance/error bars and no systematic hyperparameter sweeps to bound the effect size.

**Questions:**

- Forget-set design: How would you construct scalable, principled forget sets without manual filtering or referencing eval distributions (e.g., HumanEval)? Can you provide automated selection criteria?

- Instruction following: Please report MT-Bench, AlpacaEval, or similar pre/post-F2F to quantify conversational and instruction adherence

---

> ### Author Response · Authors · 2025-11-18
> **Response to Reviewer - N1qk  [1]**
>
> We thank reviewer N1qK for the thoughtful review. We appreciate your time to assess our work, including both the strengths you highlighted and the concerns you raised. We address each of these points below.
>
> We update the main manuscript in **red** based on the suggestions
>
> **W-1 : Regarding Feasibility of “forgetting general data” & Suggestion to design the selection forget set automatically | Q-1 : Can you provide automated selection criteria?**
>
> Through many experiments we found out that even when the forget set is constructed using BC-select and BC-mixed, the F2F consistently improves the performance. This suggest that F2F does not really need perfectly curated forget set.
> Additionally, retain set is there to balance the model’s domain specific knowledge even if there is any potential issues in forget set.
>
> Having said that, as suggested by the reviewer, we provide automated forget set selection procedure by computing the embedding of the target domain and the broader samples, and we can find the similarity using Cosine distance to get the target domain samples.
>
> We encode each data point $x$ with a Transformer :
>
> $$
> h_x = f_{\theta}(x),
> $$
>
> and define the target-domain centroid as :
>
> $$
> c_T = \frac{1}{|D_T|} \sum_{x' \in D_T} f_{\theta}(x'),
> $$
>
> then rank samples by the cosine distance : (higher distance means lower similarity)
>
> $$
> d_{\cos}(x) = 1 - \frac{h_x^\top c_T}{\lVert h_x \rVert \, \lVert c_T \rVert}.
> $$
>
> We have evaluated this and added the evaluation result in the below table (BC-Cosine is newly added forget set selection method) and in **Table 3** in main text.
>
>
> **Model: Qwen3 0.6B**
>
> | FD | Method | MBPP | HumanEval | PubMedQA | MedMCQA | Hendrycks-MATH | GSM8K |
> | --- | --- | --- | --- | --- | --- | --- | --- |
> | BC-Select | (1) GA+GD | 30.00 | 21.34 | 61.60 | 31.44 | 39.07 | 0.26 |
> | BC-Select | (1) + Tuning | 31.60 | 42.07 | 69.60 | 45.31 | 54.11 | 15.30 |
> | BC-Select | (2) GA | 24.00 | 20.73 | 60.40 | 31.29 | 25.09 | 0.24 |
> | BC-Select | (2) + SFT | 31.60 | 40.02 | 58.80 | 40.26 | 51.20 | 14.00 |
> | BC-Mixed | (1) GA+GD | 24.20 | 20.12 | 61.80 | 30.38 | 31.47 | 0.06 |
> | BC-Mixed | (1) + Tuning | 29.90 | 40.00 | 60.20 | 23.31 | 52.00 | 13.21 |
> | BC-Mixed | (2) GA | 23.80 | 20.12 | 60.20 | 31.89 | 25.00 | 0.00 |
> | BC-Mixed | (2) + Tuning | 28.00 | 33.10 | 61.20 | 35.43 | 50.00 | 13.51 |
> | BC-Cosine | (1) UA+GD | 24.01 | 18.00 | 61.20 | 29.32 | 29.05 | 0.10 |
> | BC-Cosine | (1) + Tuning | 31.55 | 41.90 | 67.33 | 45.00 | 54.85 | 13.20 |
> | BC-Cosine | (2) GA | 22.22 | 18.23 | 58.45 | 29.00 | 25.00 | 0.00 |
> | BC-Cosine | (2) + Tuning | 29.91 | 40.50 | 57.65 | 38.75 | 48.90 | 13.07 |
> | - | (3) Baseline | 22.60 | 19.50 | 57.80 | 32.25 | 41.09 | 0.02 |
> | - | (3) + Tuning | 28.80 | 31.71 | 62.60 | 42.12 | 50.06 | 12.30 |

---

> ### Author Response · Authors · 2025-11-18
> **Response to Reviewer - N1qk [2]**
>
> **Model: LLaMA3.1 8B**
>
> | FD | Method | MBPP | HumanEval | PubMedQA | MedMCQA | Hendrycks-MATH | GSM8K |
> | --- | --- | --- | --- | --- | --- | --- | --- |
> | BC-Select | (1) GA+GD | 43.60 | 54.88 | 79.21 | 60.50 | 18.00 | 61.70 |
> | BC-Select | (1) + Tuning | 60.10 | 60.37 | 89.90 | 70.25 | 29.50 | 70.51 |
> | BC-Select | (2) GA | 22.60 | 1.20 | 75.22 | 60.20 | 10.51 | 50.91 |
> | BC-Select | (2) + Tuning | 58.66 | 57.70 | 87.00 | 67.15 | 25.70 | 67.20 |
> | BC-Mixed | (1) GA+GD | 40.50 | 52.52 | 79.50 | 59.30 | 17.00 | 51.00 |
> | BC-Mixed | (1) + Tuning | 56.20 | 55.76 | 87.61 | 70.10 | 28.81 | 65.20 |
> | BC-Mixed | (2) GA | 33.20 | 25.50 | 72.30 | 57.00 | 5.20 | 35.20 |
> | BC-Mixed | (2) + Tuning | 52.30 | 40.90 | 86.90 | 61.20 | 23.01 | 66.15 |
> | BC-Cosine | (1) GA+GD | 42.55 | 53.76 | 79.00 | 58.22 | 17.91 | 61.00 |
> | BC-Cosine | (1) + Tuning | 59.55 | 59.86 | 85.31 | 71.02 | 28.33 | 68.57 |
> | BC-Cosine | (2) GA | 20.35 | 0.90 | 73.05 | 58.99 | 9.32 | 50.00 |
> | BC-Cosine | (2) + Tuning | 57.76 | 57.55 | 85.31 | 66.00 | 27.01 | 67.10 |
> | - | (3) Baseline | 49.00 | 33.54 | 75.20 | 59.43 | 19.90 | 62.85 |
> | - | (3) + Tuning | 56.60 | 56.71 | 85.31 | 64.20 | 25.51 | 66.70 |
>
> **Model: LLaMA2 13B**
>
> | FD | Method | MBPP | HumanEval | PubMedQA | MedMCQA | Hendrycks-MATH | GSM8K |
> | --- | --- | --- | --- | --- | --- | --- | --- |
> | BC-Select | (1) GA+GD | 27.22 | 0.60 | 74.90 | 38.68 | 29.00 | 5.10 |
> | BC-Select | (1) + Tuning | 50.31 | 46.15 | 90.11 | 60.10 | 51.50 | 21.50 |
> | BC-Select | (2) GA | 0.00 | 25.50 | 70.00 | 36.51 | 24.35 | 2.00 |
> | BC-Select | (2) + Tuning | 45.01 | 44.70 | 89.33 | 57.93 | 50.90 | 20.00 |
> | BC-Mixed | (1) GA+GD | 27.20 | 0.45 | 73.00 | 37.50 | 27.00 | 5.10 |
> | BC-Mixed | (1) + Tuning | 47.50 | 45.91 | 89.55 | 61.30 | 50.30 | 20.00 |
> | BC-Mixed | (2) GA | 0.00 | 10.00 | 65.99 | 29.55 | 23.55 | 1.05 |
> | BC-Mixed | (2) + Tuning | 39.55 | 40.01 | 87.00 | 50.60 | 47.60 | 20.00 |
> | BC-Cosine | (1) GA+GD | 25.30 | 0.52 | 73.44 | 37.62 | 29.09 | 6.30 |
> | BC-Cosine | (1) + Tuning | 48.91 | 43.30 | 89.00 | 58.42 | 50.33 | 18.03 |
> | BC-Cosine | (2) GA | 0.00 | 23.33 | 69.55 | 36.22 | 24.00 | 0.92 |
> | BC-Cosine | (2) + Tuning | 44.33 | 42.05 | 89.10 | 60.99 | 49.62 | 19.52 |
> | - | (3) Baseline | 27.22 | 0.60 | 75.20 | 38.68 | 27.50 | 2.00 |
> | - | (3) + Tuning | 37.01 | 40.21 | 86.30 | 54.00 | 37.09 | 16.30 |
>
> **W-2 & Q-2 : Regarding “Risk to general capabilities / conversational behavior”**
>
> We understand the concern but the base models **are not all** instruct variants. We evaluated F2F on different base models - instruct and non-instruct models (Section 3.3 main text). Additionally, F2F makes the model adapt to a certain domain without effecting the other capabilities (such as ability to chat) as it does not reply on any instruction/reasoning ability specific loss.
> However, to address the concern, we now added this explanation in Section 3.3. And to address the concern we also added Alpaca-Eval evaluation in Table-5 and below :
>
> | Model | Method | ARC-E | ARC-C | HellaSwag | Winogrande | PIQA | BoolQ | Alpaca-Eval (LC Win Rate) |
> | --- | --- | --- | --- | --- | --- | --- | --- | --- |
> | Qwen 0.6B | (1) Base Model | 68.00 | 32.50 | 45.00 | 59.00 | 67.50 | 67.50 | 28.58 |
> | Qwen 0.6B | (1) + SFT | 68.50 (↑0.5) | 36.50 (↑4.0) | 44.50 (↓0.5) | 58.50 (↓0.5) | 65.00 (↓2.5) | 71.00 (↑3.5) | 27.65 (↓0.9) |
> | Qwen 0.6B | (2) Unl₍GA+GD₎ | 67.50 (↓0.5) | 36.00 (↑3.5) | 45.50 (↑0.5) | 61.00 (↑2.0) | 68.00 (↑0.5) | 72.00 (↑4.5) | 28.38 (↓0.2) |
> | Qwen 0.6B | (2) + SFT | 68.50 (↑0.5) | 37.00 (↑4.5) | 45.00 (0) | 57.50 (↓1.5) | 64.00 (↓3.5) | 73.50 (↑6.0) | 29.01 (↑0.4) |
> | Qwen 0.6B | (3) Unl₍GA₎ | 67.50 (↓0.5) | 32.50 (0) | 59.00 (↑14.0) | 44.50 (↓14.5) | 66.50 (↓1.0) | 75.00 (↑7.5) | 29.59 (↓1.1) |
> | Qwen 0.6B | (3) + SFT | 69.00 (↑1.0) | 35.00 (↑2.5) | 45.00 (0) | 63.50 (↑4.5) | 58.50 (↓9.0) | 73.50 (↑6.0) | 27.78 (↓0.7) |
> | LLaMA 8B | (1) Base Model | 82.50 | 52.50 | 55.00 | 80.00 | 79.50 | 87.50 | 30.22 |
> | LLaMA 8B | (1) + SFT | 83.00 (↑0.5) | 56.50 (↑4.0) | 51.50 (↓3.5) | 74.30 (↓5.7) | 76.00 (↓3.5) | 89.00 (↑1.5) | 30.32 (↑0.1) |
> | LLaMA 8B | (2) Unl₍GA+GD₎ | 82.00 (↓0.5) | 56.00 (↑3.5) | 55.50 (↑0.5) | 82.00 (↑2.0) | 80.50 (↑1.0) | 90.50 (↑3.0) | 30.22 (0.0) |
> | LLaMA 8B | (2) + SFT | 83.00 (↑0.5) | 59.00 (↑6.5) | 55.00 (0) | 79.00 (↓1.0) | 78.50 (↓1.0) | 91.00 (↑3.5) | 30.55 (↑0.3) |
> | LLaMA 8B | (3) Unl₍GA₎ | 82.00 (↓0.5) | 52.50 (0) | 69.50 (↑14.5) | 65.00 (↓15.0) | 78.50 (↓1.0) | 86.50 (↓1.0) | 30.12 (↓0.1) |
> | LLaMA 8B | (3) + SFT | 84.00 (↑1.5) | 55.00 (↑2.5) | 68.00 (↑13.0) | 81.00 (↑1.0) | 77.00 (↓2.5) | 90.00 (↑2.5) | 30.42 (↑0.2) |

---

> ### Author Response · Authors · 2025-11-18
> **Response to Reviewer - N1qk [3]**
>
> **W-3 : Regarding “SFT sensitivity / insufficient controls”**
>
> We agree that understanding the robustness of F2F to SFT hyper-parameters is important. We already include several ablations that probe F2F-specific important factors: forget-set quality (BC-Select vs BC-Mixed) and size, retain-set size, and GA/GD weighting (λ, σ).
>
> In addition to that and to address the concern, we have added the performance analysis for varying learning rate for both unlearning and fine-tuning in F2F and also ran multi-seed experiments on LLaMA-3.1-8B and Qwen 2.5 7B for the coding and medical domains (3 seeds). Added in supplementary (Sections B and C in red)
>
> | Unlearning Learning Rate | PubMedQA | MedMCQA | Fine-Tuning Learning Rate | PubMedQA | MedMCQA |
> | --- | --- | --- | --- | --- | --- |
> | 5 × 10⁻⁶ | 73.91 | 48.02 | 5 × 10⁻⁶ | 84.20 | 63.00 |
> | 1 × 10⁻⁵ | 75.20 | 51.22 | 1 × 10⁻⁵ | 85.22 | 65.05 |
> | 1.5 × 10⁻⁵ | 76.99 | 54.81 | 1.5 × 10⁻⁵ | 86.71 | 67.82 |
> | 2 × 10⁻⁵ | 76.81 | 58.35 | 2 × 10⁻⁵ | 89.90 | 70.25 |
> | 2.5 × 10⁻⁵ | 79.05 | 59.21 | 2.5 × 10⁻⁵ | 89.85 | 67.09 |
> | 3 × 10⁻⁵ | 79.21 | 60.50 | 3 × 10⁻⁵ | 89.75 | 69.91 |
> | 4 × 10⁻⁵ | 78.00 | 60.33 | 4 × 10⁻⁵ | 88.00 | 69.15 |
>
> | Model | Domain | Metric | Base | SFT (3 seeds) | F2F (3 seeds) | Δ (F2F–SFT) |
> | --- | --- | --- | --- | --- | --- | --- |
> | **Coding** |  |  |  |  |  |  |
> | Qwen 2.5 7B | Coding | HumanEval pass@1 (%) | 40.20 | 44.35 ± 0.004 | 53.70 ± 0.005 | +9.4 |
> | Qwen 2.5 7B | Coding | MBPP pass@1 (%) | 65.85 | 72.90 ± 0.007 | 76.25 ± 0.004 | +3.4 |
> | LLaMA-3.1 8B | Coding | HumanEval pass@1 (%) | 33.54 | 57.37 ± 0.004 | 61.59 ± 0.005 | +4.2 |
> | LLaMA-3.1 8B | Coding | MBPP pass@1 (%) | 49.00 | 58.95 ± 0.005 | 60.73 ± 0.003 | +1.8 |
> | **Medical** |  |  |  |  |  |  |
> | Qwen 2.5 7B | Medical | PubMedQA acc. (%) | 73.00 | 81.52 ± 0.012 | 85.00 ± 0.007 | +3.5 |
> | Qwen 2.5 7B | Medical | MedMCQA acc. (%) | 56.23 | 65.35 ± 0.008 | 69.35 ± 0.002 | +4.0 |
> | LLaMA-3.1 8B | Medical | PubMedQA acc. (%) | 75.20 | 89.51 ± 0.006 | 91.35 ± 0.009 | +1.8 |
> | LLaMA-3.1 8B | Medical | MedMCQA acc. (%) | 59.43 | 64.55 ± 0.001 | 70.55 ± 0.001 | +6.0 |

---

> ### Author Response · Authors · 2025-11-27
>
> Dear Reviewer N1qK,
>
> We hope you’re doing well. We wanted to kindly check if there are any updates or if additional clarification is needed from our side. We understand everyone’s busy, but we’d really appreciate any feedback or guidance on the current status of our rebuttal.
>
> Thank you very much for your time and effort reviewing our work.

---

### Author Response · Authors · 2025-11-19
**Global Response : Additional Experiments and Updated PDF (Summarised)**

We thank all the reviewers for taking the time to review the paper and give helpful feedbacks. We updated our paper following the suggestions.

The new updates are :
**(1)** Clear motivation behind choosing the models for Table-1 and update the labels,
**(2)** Introducing automated forget set selection method : BC-Cosine
**(3)** BC-Cosine is added and evaluated in Table - 3,
**(4)** Explanation added for Gemma-2B's performance,
**(5)** Added new experiments for Qwen 2.5 models in Table-4 @supp,
**(6)** Added Alpaca-Eval (LC Win Rate) in Table-5,
**(8)** Ablation with varying learning rate both for unlearning and finetuning,
**(7)** Evaluation with different seeds in Table-9

**We hope these additional experiments and updates clarify the concerns of the reviewers. If there are any further questions, please let us know.**

---

### Meta-Review · Area_Chair_2zvi · 2026-01-06

**Summary:**

The paper provides a technique for LLM fine-tuning. The main insight is that for domain specific tasks, some information in the base model should be unlearned. The authors propose a two-step process of unlearning followed by fine-tuning. The reviewers appreciated the novelty of this approach, as well as the writing, resulting in a clear and easy to follow paper. In terms of the method, the main issue raised was the manual effort required to build a high quality forget set. The authors built an automated technique in the rebuttal stage that does not perform as well as the manual counterparts, but does seem to improve over the baseline. Another, less major issue, related to the technique is its sensitivity to hyperparameters (pointed out by N1qK). Here, the authors responded with additional ablation studies exploring more HP settings.
The second major issue raised is that of generalization. The reviews indicated a need for different benchmark types (e.g. not multi-choice but text generation) and additional domains (see comment by LotK). The authors provided additional experiments evaluating these issues, to some extent, in the rebuttal.

Overall, the sentiment for the paper was not very negative, but was not extremely positive. Given the added experiments in the rebuttal phase the paper is definitely in a better position, especially the added method to automatically find a “forget set”. However, the scope of the change combined with the relatively mild enthusiasm of the reviewers lead me to recommend that this paper undergoes another full round of review before it can be accepted to a conference such as ICLR.

**Reviewer Concerns:**

all concerns mentioned above were partially mitigated in the rebuttal. This being said, the nature of concerns is not a misunderstanding but a key missing piece for the paper. As such, the supporting evidence given in the rebuttal do not seem sufficient to fully mitigate the concern: A new method is given, with end-to-end tests, but on a specific use case, no ablation studies, and no proper explanation.

**Reviewer Scores:**

Its hard to say - it is somewhat borderline, but my best guess is that none of the reviewers would change their score

---

### Decision · Program_Chairs · 2026-01-26

Reject